# ADVERSARIAL LATENT FEATURE AUGMENTATION FOR FAIRNESS

## ABSTRACT

As fairness in machine learning has been increasingly important to mitigate bias in models, various methods to enhance fairness have been proposed. Among them, the data augmentation approach has shown promising results in improving fairness. However, existing data augmentation methods on either input or latent features provide limited evidence of how they discover bias and rectify it. In this paper, we propose the Adversarial Latent Feature Augmentation (ALFA) for fairness, which effectively merges adversarial attacks against fairness and data augmentation in the latent space to promote fairness. Though the adversarial perturbation against fairness has been discussed in existing literature, the effect of such adversarial perturbations has been inadequately studied only as a means to depreciate fairness. In contrast, in this paper, we point out that such perturbation can in fact be used to augment fairness. Drawing from a covariance-based fairness constraint, our method unveils a counter-intuitive relationship between adversarial attacks against fairness and enhanced model fairness upon training with the resultant perturbed latent features by hyperplane rotation. We theoretically prove that our adversarial fairness objective assuredly generates biased feature perturbation, and we validate with extensive experiments that training with adversarial features significantly improve fairness.

## 1 INTRODUCTION

Machine learning is widely used in the decision-making process for various applications such as screening employment applicants, evaluating credit scores, and predicting crime or abuse. Data imbalance in demographic information might evoke a biasedly trained model, and produce unfair results in the testing step. For this reason, fairness in machine learning has increasingly drawn attention to alleviating reluctant bias in learned models. Various methods of achieving fairness have been proposed with different strategies such as data reweighing, fair representation learning, fair optimization, and data augmentation.

In machine learning, data augmentation is a pre-processing method that increases data diversity. In classification tasks, data augmentation is mainly used for enhancing the classifier's accuracy. It both enlarges the number of training samples and modifies the training features into more challenging ones, thereby enabling the classifier to better adapt and learn from a broader range of data variations. In the same manner, data augmentation can be applied to improve the model's ability to make a fairer decision.

However, the transparency and efficacy of utilizing data augmentation in the input space (Jang et al., 2021; Rajabi & Garibay, 2022) to foster fairness are not inherently evident due to the inherent difficulties in determining how transformations impact the nonlinear decision hyperplane. This has led to the exploration of data augmentation strategies in the latent space, which facilitates a more in-depth study of the impact of augmentation since the decision hyperplane in the latent space is linear. For example, Fair-Mixup (Mroueh et al., 2021) operates under the assumption of the existence of a manifold in the latent space between two demographic groups and advocates for data generation on this manifold through interpolation. However, the manifold assumptions could potentially be overly stringent. FAAP (Wang et al., 2022b) aims to obfuscate sensitive attributes in the latent representation by perturbing latent features towards the sensitive hyperplane. The challenges arise as the perturbation is produced by Generative Adversarial Networks (GANs) model. In the tabular

dataset, GAN-based perturbation might yield unsuitable generated features not aligning on the sensitive hyperplane. The inadequate perturbation results in a decision boundary that is not sufficiently fair.

In response to the limitations of existing methods, we consider training on the perturbed latent features designed as antidote data to mitigate bias. We define the regions of unfairness where a demographic group is overestimated or underestimated, resulting in the misclassification rate being disproportionately high. We propose a novel data augmentation technique in the latent space, Adversarial Latent Feature Augmentation (ALFA), explicitly addressing the unfair regions to produce similar misclassification rates across demographic groups.

First, we define a fairness attack as feature-level adversarial perturbation against fairness constraint. We employ a covariance-based fairness constraint (Zafar et al., 2017) as an objective function for fairness attack, and provide theoretical insights on how attacks on such constraints can detrimentally impact fairness. By maximizing the covariance-based fairness constraint with the frozen pre-trained classifier, the perturbed latent features are highly biased, posing potential fairness challenges to the classifier. For example, after the adversarial perturbation, privileged groups may be pushed towards positive areas, while underprivileged ones towards negative areas, irrespective of their true labels, hence resulting in a high correlation between the sensitive attribute and the decision. During the adversarial training, we ensure that the semantic essence of the perturbed features is preserved by minimizing the Sinkhorn distance (Genevay et al., 2018) between original and perturbed features.

Second, we show that fine-tuning on such perturbed features can yield a 'rotated' decision boundary. Counter-intuitively, the perturbed features cover the unfair regions directly. Therefore, fine-tuning the perturbed features changes the decision boundary to incorporate the unfair regions, making demographic groups have proportionate misclassification rates. We illustrate this with synthetic data to visualize how adversarial perturbations can rotate the decision hyperplane in Figure 1.

To validate our approach, we conduct extensive experiments on diverse datasets, including Adult, COMPAS, German, and Drug datasets, and extend our concept to image classification on CelebA, demonstrating that our proposed method maintains accuracy while achieving group fairness.

We summarize our contribution as follows:

1. Introduced a novel latent space data augmentation method aimed at identifying and rectifying areas of unfairness in classification models.

2. Provided foundational theoretical perspectives elucidating the potential detrimental impacts of adversarial perturbations on fairness, supported by visual illustrations of the rotated decision hyperplane.

3. Through experiments on tabular datasets and an image dataset, our method consistently achieves group fairness with sacrificing minimum accuracy.

## 2 RELATED WORK

### 2.1 FAIRNESS IN MACHINE LEARNING

Diverse approaches have been proposed to secure fairness in the classification tasks. At first, Chai & Wang (2022) and Li & Liu (2022) proposed data reweighing; allocating weights for all samples according to their importance. Chai & Wang (2022) balanced the gap between demographic groups weighing error-prone samples in an adaptive way. Li & Liu (2022) adopted influence function (Koh & Liang, 2017) to evaluate individual sample's importance in affecting prediction. Wang et al. (2022a) also utilized the influence function as a constraint to prune influential data samples. Similarly, Zafar et al. (2017) and Wu et al. (2019) developed a fairness constraint adopting covariance between sensitive attribute and classifier, and extending the constraint having convexity.

Some approaches use data augmentation to improve fairness. Jang et al. (2021) and Rajabi & Garibay (2022) generated new fair data using VAE and GAN, respectively. Hsu et al. (2022) and Zhao et al. (2020) adopted adversarial samples as data augmentation to improve accuracy and robustness, respectively. Similarly, Li et al. (2023) generated antidote data analogous to the original data but containing the opposite sensitive attribute to enhance individual fairness.

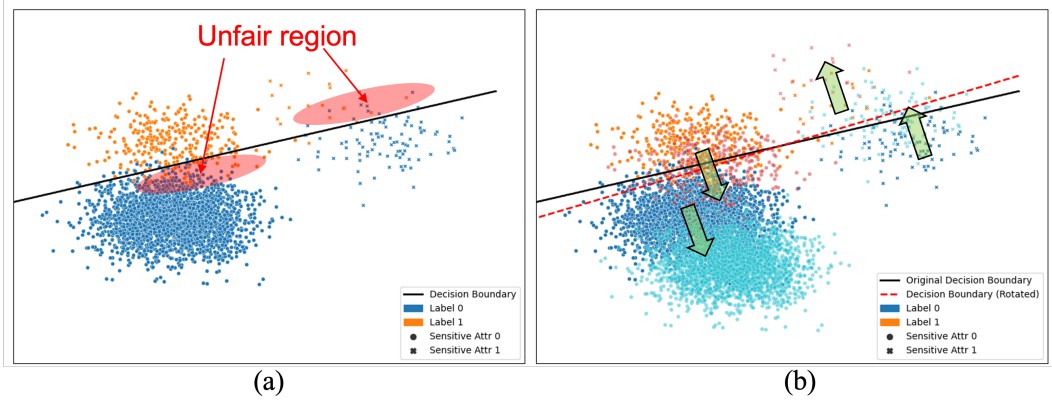

(a)                                        (b)

Figure 1: Assume that the demographic group $\{A = 1\}$ is privileged to be predicted as $Y = 1$. The misclassification rates of subgroup $\{A = 1, Y = 0\}$ and $\{A = 0, Y = 1\}$ are disproportionately high, indicated as *unfair region* in the left figure. Adversarial Latent Feature Augmentation (ALFA) generates adversarial perturbation in the latent space by the fairness constraint, moving the features towards a biased direction to make the perturbed features overlap the unfair region. Training on the perturbed features rotates the decision boundary to cover the unfair region.

Manipulating features in the latent space becomes popular. Mroueh et al. (2021) proposed to generate new data in the latent space by interpolation between latent features from different sensitive groups to optimize fairness constraints. Wang et al. (2022b) suggested adversarial perturbation on the latent features towards the sensitive hyperplane which predicts the demographic group. Sun et al. (2023) disentangle the latent feature into the sensitive feature and non-sensitive feature and obfuscate the sensitive feature only. Mao et al. (2023) fine-tune the pre-trained classifier by training the last layer with the balanced latent features under the designated fairness constraint.

In contrast, there exist attempts to attack fairness. Koh et al. (2018) suggested attacking anomaly detectors by blending perturbed data with the natural data and by optimizing influence-based gradient ascent. Mehrabi et al. (2021) extended the idea of (Koh et al., 2018) combining the fairness constraint suggested by Zafar et al. (2017). Similarly, Solans et al. (2020) developed a gradient-based poisoning attack on algorithmic fairness. Chhabra et al. (2022) proposed a fairness attack and defense framework in terms of unsupervised learning and fair clustering.

## 3 PROPOSED METHOD

**Motivation.** We use *Demographic Parity* (DP) and *Equalized Odds* (EOd) as criteria for group fairness. DP requires independence between the predicted outcome and the sensitive information $A \in \{0, 1\}$, $P(\hat{Y}|A = 0) = P(\hat{Y}|A = 1)$, i.e. $\hat{Y} \perp\!\!\!\perp A$. However, the usefulness of DP is limited to cases where there exists a correlation between $Y$ and $A$ such that $Y \not\perp\!\!\!\perp A$. EOd overcomes the limitation of DP by conditioning the metric on the ground truth $Y$, i.e. $P(\hat{Y}|A = 1, Y = y) = P(\hat{Y}|A = 0, Y = y), \forall y \in \{0, 1\}$.

In other words, EOd implies the gap of misclassification rate between two demographic groups is zero for each true label. In general, when a classifier is biased, the misclassification occurs in a particular region. For example, if instances in the privileged group have more chance to be predicted as positive, i.e. $P(\hat{Y}|A = 1) \geq P(\hat{Y}|A = 0)$, the false positive rate of group $\{A = 1\}$ and the false negative rate of a group $\{A = 0\}$ are disproportionately higher, i.e. $P(\hat{Y} = 1|A = 1, Y = 0) \geq P(\hat{Y} = 1|A = 0, Y = 0)$ and $P(\hat{Y} = 0|A = 0, Y = 1) \geq P(\hat{Y} = 0|A = 0, Y = 1)$.

As shown in Figure 1, the *unfair regions* indicate the misclassified samples reflecting the biased classifier. The proposed method aims to automatically discover the unfair regions generate perturbed samples directly covering the unfair regions with over/underestimated demographic groups for each

label, by attacking the fairness constraint. Training on the perturbed latent features produces a 'rotated' decision boundary pruning the misclassification rate of biased subgroups.

## 3.1 FAIRNESS ATTACK

In this section, we adopt an objective function suggested in (Zafar et al., 2017) for fairness attack, $\mathcal{L}_{fair}$. Zafar et al. (2017) suggested measurement for disparate impact using a covariance between the sensitive attribute $a$ and the signed distance $d_\theta$ from $\boldsymbol{x}$ to the decision boundary, i.e. $Cov(a, d_\theta) \approx 0$ means fair where the signed distance $d_\theta$ obtained by the logit (inverse sigmoid) function from the predicted probability $\hat{y}$, i.e. $d_\theta = \sigma^{-1}(\hat{y})$. Contrary to (Zafar et al., 2017), we maximize the covariance between the sensitive feature and the signed distance between the perturbed feature and the decision boundary of the pre-trained classifier. Therefore, the fairness constraint $\mathcal{L}_{fair}$ is defined

$$\mathcal{L}_{fair} = |Cov(a, \sigma^{-1}(\hat{y}))| = |Cov(a, g(\boldsymbol{z} + \delta))| \tag{1}$$

where the overall model consists of an encoder $f$ and linear classifier $g$ such that $\hat{y} = g(f(\boldsymbol{x})) = g(\boldsymbol{z})$, $\boldsymbol{x} \in \mathbb{R}^{N \times d_{input}}$ is the input, $\boldsymbol{z} \in \mathbb{R}^{N \times d}$ is the latent feature, $\boldsymbol{\delta} \in \mathbb{R}^{N \times d}$ is the perturbation, $N$ is the number of instances, and $d$ is the dimension of latent feature. Let $\tilde{\boldsymbol{z}} = \boldsymbol{z} + \delta$ and $d_i = g(\tilde{\boldsymbol{z}}_i)$, then Eq.1 becomes

$$\begin{aligned}
\mathcal{L}_{fair} &= |Cov(a, g(\tilde{\boldsymbol{z}}))| \\
&= \left| \mathbb{E}\big[(a - \bar{a})\big(g(\tilde{\boldsymbol{z}}) - \mathbb{E}[g(\tilde{\boldsymbol{z}})]\big)\big] \right| \\
&\approx \frac{1}{N_p} \left| \sum_{i=1}^{N_p} (a_i - \bar{a})(d_i - \bar{d}) \right|.
\end{aligned}$$

where $N_p$ is the number of target samples and $\bar{d}$ is the mean of all $d_i$. In the fairness attack, we adopt *upsampling strategy* selecting the same size of samples from each subgroup as a attacking target such that $N_p \approx 4 \cdot \max(n_{00}, n_{01}, n_{10}, n_{11})$, where $n_{ay}$ means the number of samples for each sensitive subset for given $y$ such that $n_{ay} = |S_{ay}|$, $S_{ay} = \{i | a_i = a, y_i = y\}$, $a \in \{0, 1\}$ and $y \in \{0, 1\}$.

**Proposition 3.1.** $\mathcal{L}_{fair}$ *is proportional to the mean signed distance gap ($\Delta d_{dp}$) between two sensitive attribute groups, and the sum of the mean signed distance gap ($\Delta d_{eod,y}$) between the sensitive groups for each ground truth label $y \in \{0, 1\}$,*

$$\begin{aligned}
\mathcal{L}_{fair} &= \frac{1}{4} \Delta d_{dp} \\
&= \frac{1}{8}\Big[ \Delta d_{eod,1} + \Delta d_{eod,0} \Big],
\end{aligned}$$

*where*

$$\Delta d_{dp} = \left| \frac{1}{n_1} \sum_{i \in S_1} d_i - \frac{1}{n_0} \sum_{j \in S_0} d_j \right|$$

$$\Delta d_{eod,1} = \left| \frac{1}{n_{11}} \sum_{i \in S_{11}} d_i - \frac{1}{n_{01}} \sum_{j \in S_{01}} d_j \right|$$

$$\Delta d_{eod,0} = \left| \frac{1}{n_{10}} \sum_{i \in S_{10}} d_i - \frac{1}{n_{00}} \sum_{j \in S_{00}} d_j \right|$$

The detailed proof for Proposition 3.1 is in Appendix C.

A positive covariance between two variables indicates that they tend to increase or decrease together, while a negative covariance means an inverse relationship. A fairness attack aims to maximize the covariance to make the sensitive attribute significantly affect the decision of the given classifier. Instead of $|Cov(a, g(\tilde{\boldsymbol{z}}))|$ in $\mathcal{L}_{fair}$, we follow the sign of covariance ($Cov(a, y)$) of the clean dataset to determine $\mathcal{L}_{fair}$ in advance of the training to help the attack-based augmented data effectively exacerbate the fairness for the given classifier,

$$\mathcal{L}_{fair} = \begin{cases} Cov(a, g(\tilde{\boldsymbol{z}})) & \text{if } Cov(a, y) \geq 0 \\ -Cov(a, g(\tilde{\boldsymbol{z}})) & \text{if } Cov(a, y) < 0. \end{cases} \tag{2}$$

---

**Algorithm 1** Adversarial Latent Feature Augmentation

---

**Require:** Clean dataset $(\boldsymbol{X}_c, \boldsymbol{Y}_c)$, hyperparameter $\alpha$, $\epsilon$ and $\lambda$, the number of epochs $T$, pretrained encoder $f$ and classifier $g$.

**Ensure:** Fair classifier $g_\theta$

   Obtain $(\boldsymbol{X}_c', \boldsymbol{Y}_c')$ by balanced upsampling for $(\boldsymbol{X}_c, \boldsymbol{Y}_c)$.

   Obtain latent feature set $(\boldsymbol{Z}, \boldsymbol{Y}_c')$ where $\boldsymbol{Z} = f(\boldsymbol{X}_c')$

   Fairness attack to obtain the latent perturbation $\delta^* = \arg\max_{\|\boldsymbol{\delta}\|_\infty \leq \epsilon}\big(\mathcal{L}_{fair} - \alpha D(\boldsymbol{z}, \boldsymbol{z} + \boldsymbol{\delta})\big)$, $\forall \boldsymbol{z} \in \boldsymbol{Z}$

   **for** $i = 1, \cdots, T$ **do**

      Fine-tune the classifier $g$ with the adversarial latent feature $\tilde{z} = \boldsymbol{z} + \boldsymbol{\delta^*}$

      $\theta \leftarrow \theta^* = \arg\min_\theta \sum_{i=1}^{N_p}\Big((1-\lambda)\mathcal{L}_{ce}(g(\boldsymbol{z}_i), y_i, \theta) + \lambda\mathcal{L}_{ce}(g(\boldsymbol{z}_i + \boldsymbol{\delta}_i^*), y_i, \theta)\Big)$

   **end for**

---

In this way, we observe in Table 2 in Appendix A that the consequent sign of $Cov(a, \hat{y})$ also follows the sign of clean dataset.

**Theorem 3.2.** *If the cardinalities of subgroups $S_{ay} = \{i | a_i = a, y_i = y\}$, $a \in \{0, 1\}, y \in \{0, 1\}$ are equal, $\mathcal{L}_{fair}$ is the lower bound of $\Delta DP$ and $\Delta EOd$ when we approximate the logit (inverse sigmoid) function as a piecewise linear function with $m$ segments s.t $d = f_k(\hat{y}) = a_k\hat{y} + b_k$ for $k \in \{1, 2, \cdots, m\}$, $m > 1, m \in \mathbb{N}$, and $a_{max} = \max(a_1, \cdots, a_k)$. Then,*

$$\mathcal{L}_{fair} \leq \frac{1}{4}\big(a_{max}\Delta DP + C\big), \tag{3}$$

$$\mathcal{L}_{fair} \leq \frac{1}{8}\big(a_{max}\Delta EOd + C_0 + C_1\big). \tag{4}$$

$C = \frac{2}{N}\sum_{k=1}^m (n_1^{(k)} - n_0^{(k)})b_k$, and $C_a = \frac{4}{N}\sum_{k=1}^m (n_1^{(k)} - n_0^{(k)})b_k$ are constants where $n_a^{(k)}$ is the number of samples in $k$-th segment for $a \in \{0, 1\}$.

We set $\beta \leq \hat{y} \leq 1 - \beta$ when we compute the signed distance to avoid $d_\theta = \ln\big(\frac{\hat{y}}{1-\hat{y}}\big) \rightarrow (\infty \text{ or } -\infty)$ so that $a_{\max} \leq \frac{m}{1-2\beta}\big[\text{logit}(\beta + \frac{1-2\beta}{m}) - \text{logit}(\beta)\big]$, theoretically. In this work, we set $\beta = 1e{-}7$ and $m = 10$ for all experiments. The detailed proof for Theorem 3.2 is in Appendix D.

We empirically verify that the naive logit function is feasible as well and effectively attacks the fairness in terms of $\Delta DP$ and $\Delta EOd$. However, the upper bound of $\mathcal{L}_{fair}$ with the naive logit function is not fully supported mathematically, while the piecewise linear logit function can be proved as Appendix . Moreover, there's no significant difference in the fairness performances between the naive logit function and its piecewise linear approximation. We choose the piecewise linear function to ensure the upper bound of $\mathcal{L}_{fair}$.

**Insights from Theorem 3.2.**  We randomly choose an equal number of samples to attack for each subset, i.e. $\frac{N_p}{4} = n_{00} = n_{01} = n_{10} = n_{11}$ to satisfy the condition in Theorem 3.2. Consequently, since $\mathcal{L}_{fair}$ is the lower bound of $\Delta DP$ and $\Delta EOd$, we can attack fairness by maximizing $\mathcal{L}_{fair}$ as the perturbed latent features produce unfair prediction with high $\Delta DP$ and $\Delta EOd$ on given pre-trained classifier.

## 3.2  SINKHORN DISTANCE

The goal of an adversarial attack is to lead a classifier to predict poor results on perturbed samples while maintaining the distribution of given data to keep it semantically meaningful. In order to effectively attack the classifier, we adopt the Wasserstein Distance (Arjovsky et al., 2017) to minimize the statistical distance between $\boldsymbol{z}$ and $\tilde{\boldsymbol{z}}$, i.e. $D(\boldsymbol{z}, \tilde{\boldsymbol{z}})$. Wasserstein distance is a powerful tool for measuring the statistical distance between two probability distributions and is sensitive to small perturbations. One drawback of Wasserstein distance is its burden on computational cost. However, a faster and more accurate algorithm is developed to approximate the Wasserstein distance using Sinkhorn iteration, Sinkhorn distance (Genevay et al., 2018). Sinkhorn Distance is an approximate

entropy regularized Wasserstein distance using the Sinkhorn algorithm measuring the distance between two probability distributions in terms of optimal transport problem. A detailed explanation of Sinkhorn distance is in Appendix B.

### 3.3 ADVERSARIAL LATENT FEATURE AUGMENTATION

We propose a novel data augmentation technique in the latent space, *Adversarial Latent Feature Augmentation* (ALFA) to mitigate the bias in the binary classification. We pre-train the encoder and classifier by Empirical Risk Minimization with binary cross entropy loss $\mathcal{L}_{ce}$

$$\min_{\theta} \frac{1}{N} \sum_{i=1}^{N} \mathcal{L}_{ce}(g(f(\boldsymbol{x}_i)), y_i).$$

where $\boldsymbol{x}_i \in \mathbb{R}^{N \times d_{\text{input}}}$ is the input and $y_i \in \{0, 1\}$ is the class label. The trained classifier is potentially biased to the particular sensitive attribute due to the imbalance in the dataset. As shown in Figure 1, unfair regions are identified for each label caused by a given classifier which we aim to cover by introducing the perturbed latent features having corresponding labels with the over/underestimated demographic group.

The adversarial latent features are generated by the fairness attack while maintaining their distributional similarity by the Sinkhorn distance. During the attacking step, parameters of both encoder $f$ and linear classifier $g$ are frozen. The direction and magnitude of perturbation are determined by the fairness attack introduced in Section 3.1 and 3.2,

$$\max_{\|\boldsymbol{\delta}\|_{\infty} \leq \epsilon} \Big( \mathcal{L}_{fair} - \alpha D(\boldsymbol{z}, \boldsymbol{z} + \boldsymbol{\delta}) \Big). \tag{5}$$

where $\alpha$ and $\epsilon$ are hyperparatmers. The Sinkhorn distance term is obtained by batch-wise computation.

At last, both the original and adversarial latent features are trained by fine-tuning where the encoder $f$ is still frozen, and only the parameters in the linear classifier $g$ are updated. The objective function for the fine-tuning is

$$\min_{\theta} \frac{1}{N_p} \sum_{i=1}^{N_p} \Big( (1 - \lambda) \mathcal{L}_{ce}(g(\boldsymbol{z}_i), y_i, \theta) + \lambda \mathcal{L}_{ce}(g(\boldsymbol{z}_i + \boldsymbol{\delta}_i^*), y_i, \theta) \Big). \tag{6}$$

In the Neural Networks, the encoder and the last layer are easily defined. However, in the Logistic Regression, there's no encoder is defined. As a special case, in the Logistic Regression, the linear classifier is pre-trained in the same manner to produce adversarial samples and trained again with our data augmentation, while the perturbation is conducted on the input space. The detailed algorithm is introduced in Algorithm 1.

## 4 EXPERIMENTAL DETAIL

### 4.1 DATASET

In this paper, we use four different tabular datasets Adult, COMPAS, German, and Drug. Also CelebA dataset is used for verify the performance of the proposed method in image classification. All the tabular datasets are split into 60:20:20 for train, validation, and test subset, respectively. The detailed description of datasets is in Appendix H.

### 4.2 EXPERIMENTAL SETUP

To verify our approach, we follow the existing data pre-processing, (Mroueh et al., 2021) for the Adult and CelebA dataset, and (Mehrabi et al., 2021) for other datasets. We apply our method to two base classifiers for tabular datasets, Logistic Regression and Multilayer Perceptron (MLP) with ReLU activation function and two hidden layers of 128 dimensions. For the CelebA dataset, we use ResNet18 (He et al., 2016) as a baseline. During the pre-training, we choose the best parameter

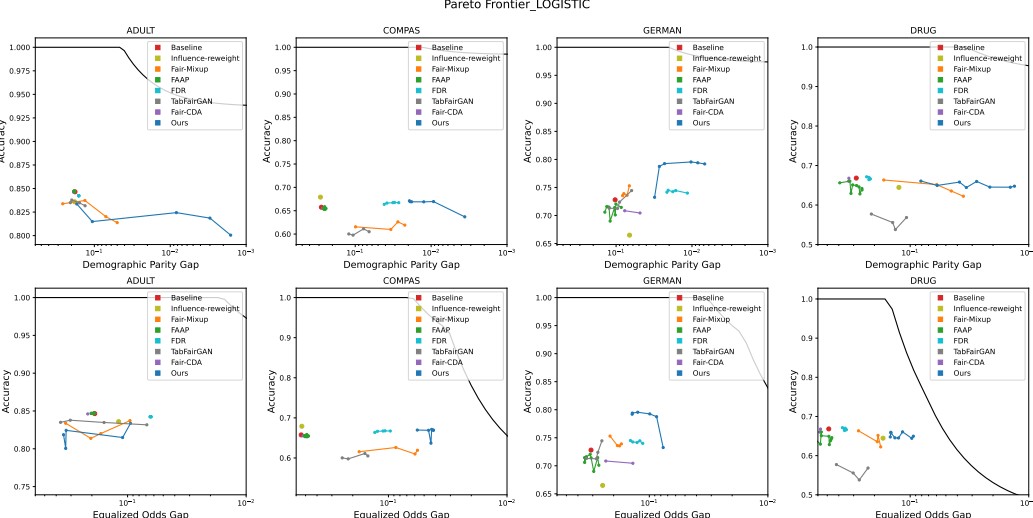

Figure 2: Pareto Frontier for Logistic Regression.

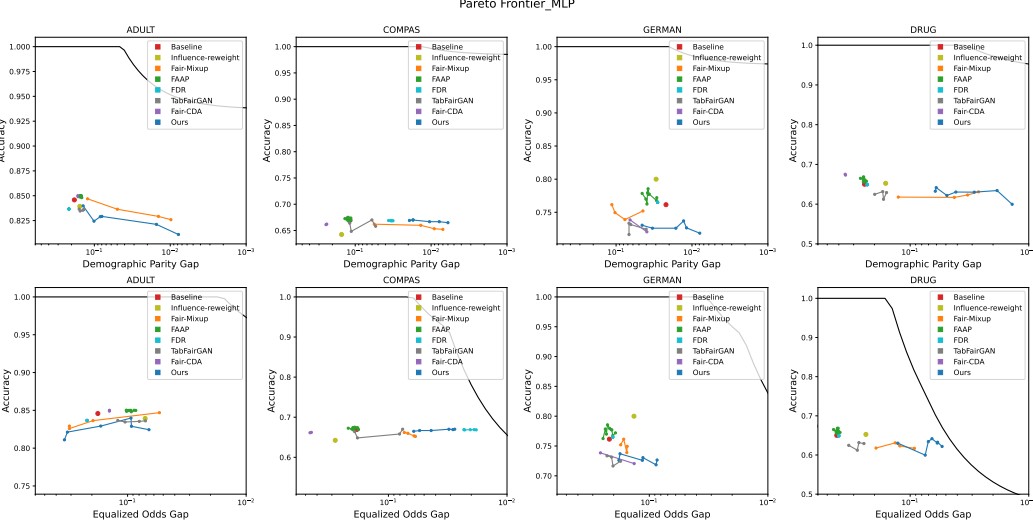

Figure 3: Pareto Frontier for MLP.

when the validation accuracy is the highest across the total epochs. In the attacking step and fine-tuning, the batch size remains the same as the pre-training. We replace the last layer with the newly initialized layer for fine-tuning, while the parameters for the encoder are frozen. The different learning rates are used in each step, Adam optimizer with learning rate $1e-3$ in pre-training and fine-tuning, Adam optimizer with learning rate 0.1 in adversarial attack. All hyperparameters are carefully tuned with the validation set. The details are provided in the Appendix E. For a fair comparison, we train each case 10 times and report the mean and the standard deviation for tabular datasets.

## 4.3 COMPARISON

To verify the ability of our method to improve fairness, we compare the experimental results of other approaches using data augmentation for fairness, data reweighing, or data manipulation in the latent space such as Fair-Mixup (Mroueh et al., 2021), TabFairGAN (Rajabi & Garibay, 2022), FAAP (Wang et al., 2022b), Fair-CDA (Sun et al., 2023), Influence-reweighing (Li & Liu, 2022) and FDR

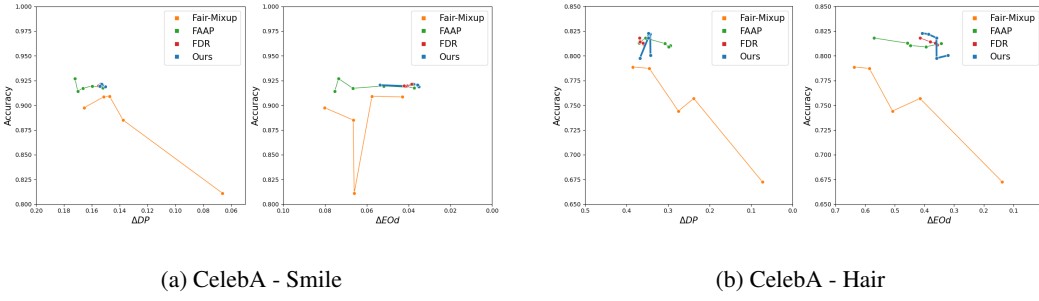

(a) CelebA - Smile          (b) CelebA - Hair

Figure 4: Experimental Results for CelebA dataset.

(Mao et al., 2023) in Pareto Frontier, Figure 2 and Figure 3. For a fair comparison, we follow their instruction such as the number of epochs and the batch size, and vary the hyperparameters for each method as explained in Appendix F.

### 4.4 RESULT ANALYSIS

#### 4.4.1 ACCURACY-FAIRNESS TRADE-OFF

The FACT Pareto Frontier (Kim et al., 2020) aims to present the trade-off between fairness metrics and accuracy and shows a model's achievable accuracy. The optimal line in FACT Pareto Frontier is based on a fairness-confusion matrix consisting of 8 values according to two sensitive attributes $a \in \{0, 1\}$, two truth labels $y \in \{0, 1\}$, and two prediction labels $\hat{y} \in \{0, 1\}$.

In the tabular dataset, ALFA shows the best fairness improvement in most cases, otherwise presents comparable performance with the best ones as shown in Figure 2, Figure 3 and Figure 4. In detail, ALFA shows an outstanding accuracy-fairness trade-off in all cases with logistic regression except $\Delta EOd$ for the Adult dataset, while still showing comparable results with the best comparison method. Similarly, the experimental results with MLP indicate that ours may not be the best in all cases such as the Adult dataset and $\Delta EOd$ in the COMPAS dataset, but is still comparable to the best $\Delta EOd$. In short, ALFA has a trade-off only in the Adult dataset, since the unfair regions might contain correctly predicted samples as well. As ALFA doesn't have an accuracy constraint in the attacking step, there's the possibility that the rotated decision boundary prioritizes unfairly classified features, while compromising the correctly predicted features.

In the visual recognition task, CelebA, ALFA shows the best accuracy-fairness trade-off compared to FDR and Fair-Mixup, while FAAP is comparable to ours. However, FAAP doesn't improve fairness in the case of tabular datasets. In conclusion, ALFA generally augments fairness, demonstrating superior or competitive results in varied datasets and tasks. The detailed experimental results for tabular and visual datasets are provided in Appendix G.

#### 4.4.2 ANALYSIS FOR THE COMPARISONS

We analyze how such approaches, FAAP, Fair-Mixup, and ALFA improve fairness on a synthetic dataset as shown in Figure 5. In FAAP, the author generates adversarial perturbation using GAN model towards the sensitive hyperplane to make the sensitive attributes not recognizable, while trying to maintain the accuracy. In the simplified form the objective function becomes,

$$\min_{\theta} \big( \mathcal{L}_{ce}(f_\theta, \boldsymbol{x} + \boldsymbol{\delta}, y) - \mathcal{L}_{ce}(f_\theta, \boldsymbol{x} + \boldsymbol{\delta}, a) \big).$$

However, in FAAP, the perturbations are not necessarily towards the sensitive hyperplane as shown in Figure 5b, especially in the tabular dataset. There could potentially be two reasons for the observed discrepancies: the variations in the population sizes of each demographic group and the possible unsuitability of GAN-based perturbation for tabular datasets. Moreover, although the perturbed samples are correctly projected to the sensitive hyperplane, it doesn't necessarily lead to the fairer classifier. In Fair-Mixup, the author uses an interpolation strategy to generate data in the manifold. However, the manifold assumptions could be too strict. Moreover, although the interpolated

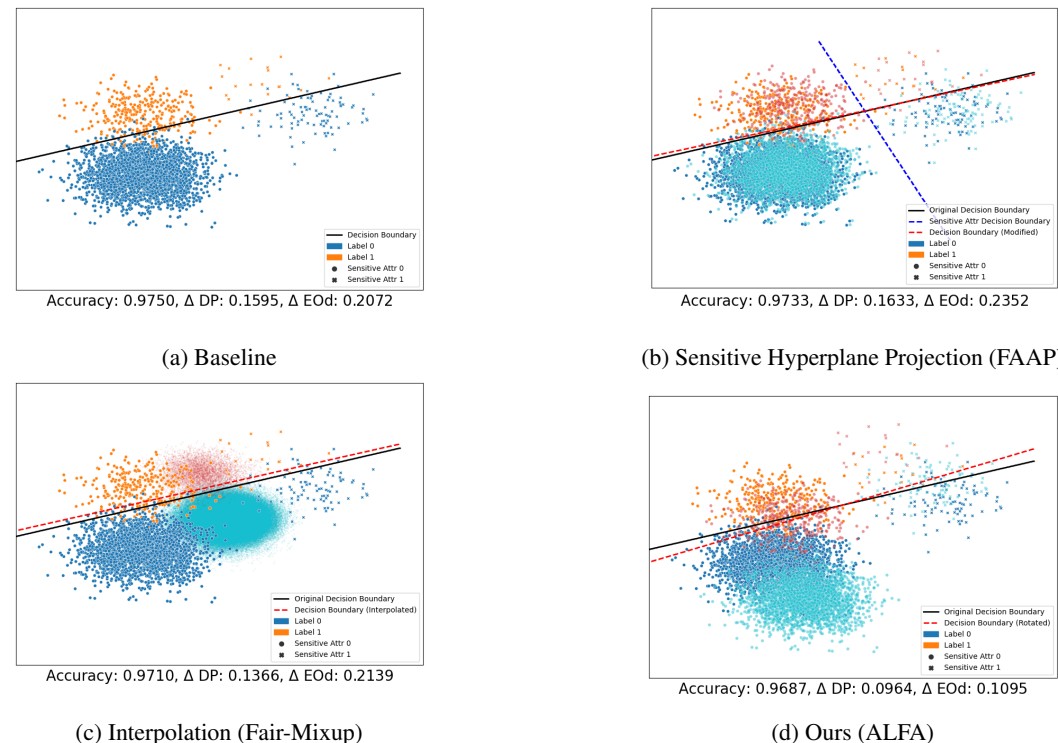

Figure 5: Comparison of each concept for synthetic unfair data. The light colors (cyan and red) indicate the manipulated feature. For example, The light colors are perturbed features in FAAP and ALFA and interpolated features in Fair-Mixup. The solid black line is the original decision boundary obtained by Logistic Regression, whereas the red dashed line is the updated decision boundary. The blue dashed line in FAAP indicates the sensitive hyperplane.

data may compensate for the imbalance in the dataset, it doesn't take into account the unfair regions, where the misclassification rates are disproportionately high, as shown in Figure 5c.

On the other hand, as discussed in Section 3 and Figure 1, ALFA directly discovers and covers the unfair regions to rotate the classifier to become fairer.

## 5 CONCLUSION

In this research, we address the prominent issue of fairness in machine learning models, focusing on biased models emanating from data imbalance in demographic information. Through a novel approach, Adversarial Latent Feature Augmentation (ALFA), we effectively manage to identify and rectify areas of unfairness in classification models, fostering more equitable decision-making processes. ALFA ensures the production of biasedly perturbed features by a fairness attack based on covariance-based constraint. We theoretically show the fairness attack successfully produces biased latent features. Moreover, we show that fine-tuning the classifier on the perturbed samples ultimately mitigates the discrepancy in misclassification rates among different demographic groups. The counterintuitive result of how the biased samples can be remedied to alleviate bias is explained in illustration with synthetic data. Our method is validated through extensive experiments on diverse datasets. Remarkably, ALFA is successful in maintaining accuracy while achieving group fairness across all datasets, showcasing its effectiveness in promoting unbiased machine learning models. In future directions, we will further extend the proposed method to take into account fairness in multi-class classification and multi-sensitive attributes problem which is underexplored in this field.

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

## A    COVARIANCE BETWEEN THE LABEL AND THE SENSITIVE ATTRIBUTE.

Table 1: The estimated value of $Cov(a, y)$ and $Cov(a, \hat{y})$. We set the sign of $\mathcal{L}_{fair}$ the same as the covariance.

|        | $Cov(a, y)$ | $Cov(a, \hat{y})$ | $\mathcal{L}_{fair}$ |
|--------|-------------|-------------------|----------------------|
| Adult  | 0.0439      | 0.0441            | $Cov(a, g(\tilde{z}))$  |
| COMPAS | -0.0198     | -0.0194           | $-Cov(a, g(\tilde{z}))$ |
| German | 0.0210      | 0.0188            | $Cov(a, g(\tilde{z}))$  |
| Drug   | 0.0434      | 0.0401            | $Cov(a, g(\tilde{z}))$  |

## B    SINKHORN DISTANCE

Optimal transport with lowest cost is defined as $\mathcal{L}_C = \min_{P} \sum_{i,j} C_{ij} P_{ij}$, where $C$ is a *cost matrix* (2-Wasserstein Distance), and $P$ is the *coupling matrix*. Genevay et al. (2018) suggested a regularized optimal transport scheme which includes entropy term to secure stability and smoothness of $P$, $\mathcal{L}_C = \min_{P} \sum_{i,j} C_{ij} P_{ij} - \epsilon_s H(P)$ where $H(P) = -\sum_{ij} P_{ij} \log P_{ij}$. $\mathcal{L}_C$ can be solved by *Sinkhorn iteration*, s.t. $P_{ij} = \text{diag}(u_i) K_{ij} \text{diag}(v_j)$, and updated alternately,

$$u^{(k+1)} = \frac{a}{Kv^{(k)}},$$

$$v^{(k+1)} = \frac{b}{K^{\mathsf{T}} u^{(k+1)}},$$

where $P1 = a$, $P^{\mathsf{T}} 1 = b$, and Gibbs kernel $K_{ij} = e^{-c_{ij}/\epsilon_s}$. Therefore, the distance between clean data $x$ and perturbed data $\tilde{x}$ can be rewritten as follows,

$$D(x, \tilde{x}) = \text{Sinkhorn Distance}(x, \tilde{x})$$
$$= \min_{P(x,\tilde{x})} \sum_{i,j} C_{ij}(x, \tilde{x}) P_{ij}(x, \tilde{x}) - \epsilon_s H(P(x, \tilde{x})).$$

## C    PROOF OF PROPOSITION 3.1

*Proof.* Let $d_i = g(\tilde{z})$, $\bar{d}$ is the mean of all $d_i$ and

$$\Delta d_{dp} = \left| \frac{1}{n_1} \sum_{i \in S_1} d_i - \frac{1}{n_0} \sum_{j \in S_0} d_j \right|$$

$$\Delta d_{eod,1} = \left| \frac{1}{n_{11}} \sum_{i \in S_{11}} d_i - \frac{1}{n_{01}} \sum_{j \in S_{01}} d_j \right|$$

$$\Delta d_{eod,0} = \left| \frac{1}{n_{10}} \sum_{i \in S_{10}} d_i - \frac{1}{n_{00}} \sum_{j \in S_{00}} d_j \right|$$

where $S_a$ is a subset containing each sensitive attributes $S_a = \{i | a_i = a\}$, $a \in \{0, 1\}$, and $n_{ay}$ means the number of samples for each sensitive subset for given $y$, $S_{ay} = \{i | a_i = a, y_i = y\}$, $a \in \{0, 1\}$ and $y \in \{0, 1\}$. In our experiments, we select samples with the same size such that $\frac{N}{4} = n_{00} = n_{01} = n_{10} = n_{11}$, and $\frac{N}{2} = n_0 = n_1$ where $n_0 = n_{00} + n_{01}$, $n_1 = n_{10} + n_{11}$, and $N = n_{00} + n_{01} + n_{10} + n_{11}$.

The objective function $\mathcal{L}_{fair} = |Cov(a, g(\tilde{z}))|$ can be rewritten as

$$
\begin{aligned}
\mathcal{L}_{fair} &= \frac{1}{N}\left|\sum_{i=1}^{N}(a_i - \bar{a})(d_i - \bar{d})\right| \\
&= \frac{1}{N}\left|\sum_{i \in S_1}(1 - \bar{a})(d_i - \bar{d}) + \sum_{j \in S_0}(0 - \bar{a})(d_j - \bar{d})\right| \\
&= \frac{1}{N^2}\left|n_0 \sum_{i \in S_1}(d_i - \bar{d}) - n_1 \sum_{j \in S_0}(d_j - \bar{d})\right| \\
&= \frac{1}{N^2}\left|n_0 \sum_{i \in S_1} d_i - n_1 \sum_{j \in S_0} d_j - n_0 n_1 \bar{d} + n_0 n_1 \bar{d}\right| \\
&= \frac{n_0 n_1}{N^2}\left|\frac{1}{n_1}\sum_{i \in S_1} d_i - \frac{1}{n_0}\sum_{j \in S_0} d_j\right| \\
&= \frac{1}{4}\Delta d_{dp}.
\end{aligned}
\tag{7}
$$

Similarly, we can conditionize $\mathcal{L}_{fair}$ in terms of $y$,

$$
\begin{aligned}
\mathcal{L}_{fair} &= \frac{1}{N}\left|\sum_{i=1}^{N}(a_i - \bar{a})(d_i - \bar{d})\right| \\
&= \frac{1}{N}\left|\sum_{i \in S_{11}}(1 - \bar{a})(d_i - \bar{d}) + \sum_{j \in S_{01}}(0 - \bar{a})(d_j - \bar{d}) + \sum_{i \in S_{10}}(1 - \bar{a})(d_i - \bar{d}) + \sum_{j \in S_{00}}(0 - \bar{a})(d_j - \bar{d})\right| \\
&= \frac{1}{N}\left|\sum_{i \in S_{11}}\frac{(n_{01} + n_{00})}{N}(d_i - \bar{d}) - \sum_{j \in S_{01}}\frac{(n_{11} + n_{10})}{N}(d_j - \bar{d})\right. \\
&\quad \left. + \sum_{i \in S_{10}}\frac{(n_{01} + n_{00})}{N}(d_i - \bar{d}) - \sum_{j \in S_{00}}\frac{(n_{11} + n_{10})}{N}(d_j - \bar{d})\right| \\
&= \left|\frac{n_{11}(n_{01} + n_{00})}{N^2}\frac{1}{n_{11}}\sum_{i \in S_{11}} d_i - \frac{n_{01}(n_{11} + n_{10})}{N^2}\frac{1}{n_{01}}\sum_{j \in S_{01}} d_j\right. \\
&\quad \left. + \frac{n_{10}(n_{01} + n_{00})}{N^2}\frac{1}{n_{10}}\sum_{i \in S_{10}} d_i - \frac{n_{00}(n_{11} + n_{10})}{N^2}\frac{1}{n_{00}}\sum_{j \in S_{00}} d_j\right| \\
&= \frac{n_{11} n_0}{N^2}\left[\left|\frac{1}{n_{11}}\sum_{i \in S_{11}} d_i - \frac{1}{n_{01}}\sum_{j \in S_{01}} d_j\right| + \left|\frac{1}{n_{10}}\sum_{i \in S_{10}} d_i - \frac{1}{n_{00}}\sum_{j \in S_{00}} d_j\right|\right] \\
&= \frac{1}{8}\left[\Delta d_{eod,1} + \Delta d_{eod,0}\right]
\end{aligned}
\tag{8}
$$

$\square$

# D  PROOF OF THEOREM 3.2

*Proof.* As we fix the sign of $\mathcal{L}_{fair}$ following the sign of $Cov(a, y)$, the sign of $\Delta d_{dp}$ and $\Delta DP$ are particularly defined as

$$
\begin{aligned}
\Delta d_{dp} &= \left| \frac{1}{n_1} \sum_{i \in S_1} d_i - \frac{1}{n_0} \sum_{j \in S_0} d_j \right| \\
&= \frac{1}{n_1} \sum_{i \in S_1} d_i - \frac{1}{n_0} \sum_{j \in S_0} d_j \\
\Delta DP &= \left| \frac{1}{n_1} \sum_{i \in S_1} \hat{y}_i - \frac{1}{n_0} \sum_{j \in S_0} \hat{y}_j \right| \\
&= \frac{1}{n_1} \sum_{i \in S_1} \hat{y}_i - \frac{1}{n_0} \sum_{j \in S_0} \hat{y}_j
\end{aligned}
$$

when we assume that $Cov(a, y)$ is positive. In the negative case, the sign of $\Delta d_{dp}$ and $\Delta DP$ will be changed simultaneously.

If we assume the logit function as a piecewise linear function with $m$ segments s.t $m > 1, m \in \mathbb{N}$, and recall that $n_{dp} = n_0 = n_1$ and $\frac{N}{4} = n_{00} = n_{01} = n_{10} = n_{11}$. Let each linear function is $d = f_k(\hat{y}) = a_k \hat{y} + b_k$, $k = 1, 2, \cdots, m$. Then the $\Delta d_{dp}$ and $\Delta DP$ becomes

$$
\begin{aligned}
\Delta d_{dp} &= \frac{1}{n_1} \sum_{i \in S_1} d_i - \frac{1}{n_0} \sum_{j \in S_0} d_j \\
&= \sum_{k=1}^{m} \left[ \frac{1}{n_1} \sum_{i \in S_1^{(k)}} (a_k \hat{y}_i + b_k) - \frac{1}{n_0} \sum_{j \in S_0^{(k)}} (a_k \hat{y}_j + b_k) \right] \\
&= \frac{1}{n_{dp}} \sum_{k=1}^{m} a_k \left( \sum_{i \in S_1^{(k)}} \hat{y}_i - \sum_{j \in S_0^{(k)}} \hat{y}_j \right) \\
&\quad + \frac{1}{n_{dp}} \sum_{k=1}^{m} (n_1^{(k)} - n_0^{(k)}) b_k \\
&\le \frac{a_{\max}}{n_{dp}} \left[ \sum_{k=1}^{m} \left( \sum_{i \in S_1^{(k)}} \hat{y}_i - \sum_{j \in S_0^{(k)}} \hat{y}_j \right) \right] + C \\
&= \frac{a_{\max}}{n_{dp}} \left[ \sum_{i \in S_1} \hat{y}_i - \sum_{j \in S_0} \hat{y}_j \right] + C \\
&= a_{\max} \Delta DP + C
\end{aligned} \tag{9}
$$

where $n_a^{(k)}$ means the number of samples in $k$-th segment for $a \in \{0, 1\}$ and $C = \frac{1}{n_{dp}} \sum_{k=1}^{m} (n_1^{(k)} - n_0^{(k)}) b_k$ is a constant. Therefore, maximizing $\mathcal{L}_{fair}$ maximizes $\Delta DP$ since

$$
\mathcal{L}_{fair} = \frac{1}{4} \Delta d_{dp} \le \frac{1}{4} \left( a_{\max} \Delta DP + C \right). \tag{10}
$$

Similarly, the same proof can be applied to the relationship between $\Delta d_{eod,0}$, $\Delta d_{eod,1}$, and $\Delta EOd$ as explained in Eq. (9), such that

$$\Delta EOd = \left| \frac{1}{n_{11}} \sum_{i \in S_{11}} \hat{y}_i - \frac{1}{n_{01}} \sum_{j \in S_{01}} \hat{y}_j \right| + \left| \frac{1}{n_{10}} \sum_{i \in S_{10}} \hat{y}_i - \frac{1}{n_{00}} \sum_{j \in S_{00}} \hat{y}_j \right|$$

$$= \frac{1}{n_{eod}} \left[ \left( \sum_{i \in S_{11}} \hat{y}_i - \sum_{j \in S_{01}} \hat{y}_j \right) + \left( \sum_{i \in S_{10}} \hat{y}_i - \sum_{j \in S_{00}} \hat{y}_j \right) \right]$$

$$\Delta d_{eod,1} = \left| \frac{1}{n_{11}} \sum_{i \in S_{11}} d_i - \frac{1}{n_{01}} \sum_{j \in S_{01}} d_j \right|$$

$$= \frac{1}{n_{eod}} \left[ \sum_{i \in S_{11}} d_i - \sum_{j \in S_{01}} d_j \right] \leq \frac{a_{\max}}{n_{eod}} \left[ \sum_{i \in S_{11}} \hat{y}_i - \sum_{j \in S_{01}} \hat{y}_j \right] + C_1$$

$$\Delta d_{eod,0} = \left| \frac{1}{n_{10}} \sum_{i \in S_{10}} d_i - \frac{1}{n_{00}} \sum_{j \in S_{00}} d_j \right|$$

$$= \frac{1}{n_{eod}} \left[ \sum_{i \in S_{10}} d_i - \sum_{j \in S_{00}} d_j \right] \leq \frac{a_{\max}}{n_{eod}} \left[ \sum_{i \in S_{10}} \hat{y}_i - \sum_{j \in S_{00}} \hat{y}_j \right] + C_0$$

Therefore,

$$\mathcal{L}_{fair} = \frac{1}{8} \left[ \Delta d_{eod,1} + \Delta d_{eod,0} \right]$$

$$\leq \frac{1}{8} \left[ \frac{a_{\max}}{n_{eod}} \left[ \left( \sum_{i \in S_{11}} \hat{y}_i - \sum_{j \in S_{01}} \hat{y}_j \right) + \left( \sum_{i \in S_{10}} \hat{y}_i - \sum_{j \in S_{00}} \hat{y}_j \right) \right] + C_0 + C_1 \right]$$

$$= \frac{1}{8} \left[ a_{\max} \Delta EOd + C_0 + C_1 \right] \tag{11}$$

where $C_a = \sum_{k=1}^{m} (n_1^{(k)} - n_0^{(k)}) b_k, a \in \{0, 1\}$ are constants. $\qquad \square$

## E   HYPERPARAMETERS

Table 2: Hyperparameters for the experiments for ALFA.

| Hyperparameter | Search-range |
|---|---|
| $\epsilon$ | [0.05, 0.1, 0.5, 1.0] |
| $\lambda$ | [0.25, 0.5, 0.75, 1.0] |
| $\alpha$ | [0.0, 0.1, 1.0, 10, 100] |
| total epoch $T$ | [10, 50, 200] |
| attacking iteration | 100 |
| batch size | 128 |

## F   EXPERIMENTAL SETTINGS FOR FAIR COMPARISON

**Fair-Mixup.** Fair-Mixup is an in-processing data augmentation using interpolation on manifold between two sensitive groups. Smooth regularizers for linear interpolation on DP and EOd are as follows

$$R_{mixup}^{DP} = \int_0^1 \left| \int \langle \nabla_x f(tg(x_0) + (1-t)g(x_1)), g(x_0) - g(x_1) \rangle dP_0(x_0) dP_1(x_1) \right| dt,$$

$$R_{mixup}^{EOd} = \sum_{y \in \{0,1\}} \int_0^1 \left| \int \langle \nabla_x f(tx_0 + (1-t)x_1), x_0 - x_1 \rangle dP_0^y(x_0) dP_1^y(x_1) \right| dt,$$

where $g : \mathcal{X} \rightarrow \mathcal{Z}$ is a feature encoder. The final objective function of Fair Mixup is

$$\mathcal{L}_{mixup} = L_{acc} + \lambda R_{mixup}(f).$$

For a fair comparison, we vary the ratio of regularizer adjusting $\lambda \in \{0.1, 0.3, 0.5, 0.7\}$ for tabular datasets and $\lambda = 20$ for CelebA dataset as suggested in the released implementation.

**TabFairGAN.** It aims to produce high-quality tabular data containing the same joint distribution as the original dataset using Wasserstein GAN. The training algorithm in (Rajabi & Garibay, 2022) consists of two phases, training for accuracy (phase 1) and training for both accuracy and fairness (phase 2). In both phases, the loss function for critics $C$ adopts gradient penalty (Gulrajani et al., 2017).

$$V_c = \mathbb{E}_{\hat{\boldsymbol{x}} \sim P_g}[C(\hat{\boldsymbol{x}})] - \mathbb{E}_{\boldsymbol{x} \sim P_r}[C(\boldsymbol{x})] + \lambda_c \mathbb{E}_{\bar{\boldsymbol{x}} \sim P_g}[(\|\nabla_{\bar{\boldsymbol{x}}} C(\bar{\boldsymbol{x}})\|_2 - 1)^2]$$

The loss function for the generator differs from each phase.

$$V_G = - \mathbb{E}_{\hat{\boldsymbol{x}} \sim P_g}[C(\hat{\boldsymbol{x}})] \tag{phase 1}$$

$$V_G = - \mathbb{E}_{\hat{\boldsymbol{x}}, \hat{\boldsymbol{y}}, \hat{\boldsymbol{a}} \sim P_g}[C(\hat{\boldsymbol{x}}, \hat{\boldsymbol{y}}, \hat{\boldsymbol{a}})] - \lambda_f(\mathbb{E}_{\hat{\boldsymbol{x}}, \hat{\boldsymbol{y}}, \hat{\boldsymbol{a}} \sim P_g}[\hat{\boldsymbol{y}}|\hat{\boldsymbol{a}} = 0] - \mathbb{E}_{\hat{\boldsymbol{x}}, \hat{\boldsymbol{y}}, \hat{\boldsymbol{a}} \sim P_g}[\hat{\boldsymbol{y}}|\hat{\boldsymbol{a}} = 1]) \tag{phase 2}$$

where $\lambda_f$ is hyperparameter. We excute TabFairGAN with various $\lambda_f \in \{0.1, 0.3, 0.5, 0.7\}$ for fair comparison since the implementation uses $\lambda_f = 0.5$.

**FAAP** FAAP aims to generate a perturbation using GANs, while the generator makes perturbation and the discriminator predicts the perturbed features' sensitive attributes. In formula,

$$\mathcal{L}_D = \mathcal{L}_{ce}(D(f(\hat{x}), a))$$
$$\mathcal{L}_G^{fair} = -\mathcal{L}_D - \alpha \mathcal{H}(D(f(\hat{x}), a))$$
$$\mathcal{L}_G^T = \mathcal{L}_{ce}(g(f(\hat{x})), y)$$

where $G$ is a generator, $D$ is a discriminator, $\hat{x}$ is the perturbed samples, $\mathcal{H}$ is the entropy, $g$ is label predictor (classifier), and $f$ is an encoder. The final formulation becomes

$$\arg \max_G \min_D \mathcal{L}_{ce}(D(g(\hat{x})), a) + \alpha \mathcal{H}(D(g(\hat{x})) - \beta \mathcal{L}_G^T$$

where $g(\hat{x}) = g(x + G(x))$. As the architectures for the generator and discriminator are not provided, we set a generator as an MLP model with two hidden layers with 128 nodes, having a ReLU activation function. For the discriminator, we adopt the same network with the label predictor in each tabular dataset and image dataset. For the fair comparison, we grid search $\alpha$ and $\beta$ by $\alpha \in \{0.1, 1.0, 10\}$ and $\beta \in \{0.1, 1.0, 10\}$ since the value is not given in the original paper.

**FDR** FDR is a simple fine-tuning method, including balanced sampling in the latent features, and use fairness constraint as a objective function. In detail, the Equalized-odds-based fairness constraint is

$$fpr = \left| \frac{\sum_i p_i(1 - y_i)a_i}{\sum_i a_i} - \frac{\sum_i p_i(1 - y_i)(1 - a_i)}{\sum_i(1 - a_i)} \right|$$
$$fnr = \left| \frac{\sum_i (1 - p_i)y_i a_i}{\sum_i a_i} - \frac{\sum_i (1 - p_i)y_i(1 - a_i)}{\sum_i(1 - a_i)} \right|$$

where $p_i$ denotes the predicted probability. The final objective function for the fine-tuning is

$$\min_\theta \left[ \mathcal{L}_{ce}(g) + \alpha(fpr + fnr) \right].$$

As suggested in the original paper, we search $\alpha \in \{0.5, 1, 2, 5, 10\}$.

**Fair-CDA** Fair-CDA aims to disentangle latent features into 'sensitive feature' and 'non-sensitive feature', and obfuscate the sensitive features to obtain a fairer classifier. Fair-CDA consists of three extractor, $h$, $h_y$, and $h_a$ as

$$z_i = h(x_i), z_i^y = h_y(z_i), z_i^a = h_a(z_i)$$

$h_y$ should extract features only related to the label predictions, while $h_a$ is related to the sensitive attribute only. The regularization becomes

$$\beta(\mathcal{L}_i^y + \mathcal{L}_i^a + \mathcal{L}_i^\perp)$$

and

$$\mathcal{L}_i^y = \mathcal{L}_{ce}(g_y(z_i^y), y_i)$$
$$\mathcal{L}_i^a = \mathcal{L}_{ce}(g_a(z_i^a), a_i)$$
$$\mathcal{L}_i^\perp = \frac{\langle \nabla_{z_i}\mathcal{L}_i^y, \nabla_{z_i}\mathcal{L}_i^a \rangle^2}{\|\nabla_{z_i}\mathcal{L}_i^y\|^2 \cdot \|\nabla_{z_i}\mathcal{L}_i^a\|^2}.$$

where $g_y$ and $g_a$ are two classifier for $y$ and $a$, respectively. In stage 1 for the first 450 epochs, the objective function is

$$\frac{1}{n}\sum_{i=1}^n \mathcal{L}_i + \beta(\mathcal{L}_i^y + \mathcal{L}_i^a + \mathcal{L}_i^\perp),$$

where $\mathcal{L}_i = \mathcal{L}_{ce}(g([z_i^y, z_i^a]), y_i)$. For stage 2, Fair-CDA conducts semantic augmentation to make the sensitive features along the direction to increase the attribute loss,

$$\tilde{z}_i^a = z_i^a + \alpha_i \frac{\nabla_{z_i^a}\mathcal{L}_{ce}(g_a(z_i^a), a_i)}{\|\nabla_{z_i^a}\mathcal{L}_{ce}(g_a(z_i^a), a_i)\|}$$

Based on the obtained $\tilde{z}_i^a$ and the solution of the task model in stage 1; $\hat{g}$, obtain two loss functions for stage 2 for 50 epochs,

$$\tilde{\mathcal{L}}_i = \mathcal{L}_{ce}(g([z_i^y, \tilde{z}_i^a]), y_i)$$
$$\hat{\mathcal{L}}_i = \mathcal{L}_{ce}(g([z_i^y, \tilde{z}_i^a]), \hat{g}([z_i^y, \tilde{z}_i^a])).$$

Then, the final objective function for stage 2 becomes

$$\frac{1}{n}\sum_{i=1}^n \gamma\tilde{\mathcal{L}}_i + (1-\gamma)\hat{\mathcal{L}}_i + \beta(\mathcal{L}_i^y + \mathcal{L}_i^a + \mathcal{L}_i^\perp).$$

Fair-CDA requires five hyperparameters, perturbation size $\alpha_i$ randomly drawn by $U(0, \lambda)$ where $\lambda \in \{0, 1, 10, 100, 1000\}$. $\gamma = 0.9$ as written in the paper, and $\beta$ is the initial loss value. As the learning rate for stages 1 and 2 are not given, we grid search $\eta_1, \eta_2 \in \{0.0001, 0.001, 0.01\}$ as well as $\lambda$.

## G  DETAILS IN EXPERIMENTAL RESULTS

Result comparison for accuracy, $\Delta DP$, and $\Delta EOd$ with other fairness approaches. Blue means the best result for each dataset, and Cyan means the second best. Red means the worst case. The results are reported when the accuracy is the highest among the points in the Pareto Frontier for each method. The results show that our method is always the best or second best in improving fairness.

| Adult | Logistic Regresesion | | |
|---|---|---|---|
|  | Acc. | $\Delta DP$ | $\Delta EOd$ |
| Fair-Mixup | 0.8373±0.0221 | 0.1333±0.0117 | 0.0954±0.0116 |
| FAAP | 0.8473±0.0012 | 0.1848±0.0127 | 0.1922±0.0234 |
| FDR | 0.8425±0.0003 | 0.1608±0.0018 | 0.0635±0.0015 |
| TabFairGAN | 0.8378±0.0004 | 0.1983±0.0025 | 0.3043±0.0084 |
| Fair-CDA | 0.8461±0.0003 | 0.1794±0.0010 | 0.1870±0.0040 |
| Ours | 0.8336±0.0012 | 0.1703±0.0032 | 0.0945±0.0065 |

| Adult | MLP | | |
|---|---|---|---|
|  | Acc. | $\Delta DP$ | $\Delta EOd$ |
| Fair-Mixup | 0.8469±0.0032 | 0.1228±0.0084 | 0.0539±0.0180 |
| FAAP | 0.8502±0.0013 | 0.1501±0.0164 | 0.1025±0.0418 |
| FDR | 0.8367±0.0001 | 0.2160±0.0005 | 0.2191±0.0011 |
| TabFairGAN | 0.8366±0.0065 | 0.1586±0.0072 | 0.1209±0.0225 |
| Fair-CDA | 0.8501±0.0016 | 0.1650±0.0031 | 0.1421±0.0164 |
| Ours | 0.8398±0.0009 | 0.1405±0.0034 | 0.0936±0.0092 |

| COMPAS | Logistic Regresesion | | |
|---|---|---|---|
|  | Acc. | $\Delta DP$ | $\Delta EOd$ |
| Fair-Mixup | 0.6261±0.0255 | 0.0274±0.0176 | 0.0866±0.0284 |
| FAAP | 0.6586±0.0035 | 0.2529±0.0204 | 0.4888±0.0451 |
| FDR | 0.6676±0.0035 | 0.0304±0.0187 | 0.1069±0.0393 |
| TabFairGAN | 0.6112±0.0229 | 0.0776±0.0633 | 0.1579±0.1169 |
| Fair-CDA | 0.6589±0.0004 | 0.2712±0.0019 | 0.5260±0.0045 |
| Ours | 0.6696±0.0017 | 0.0093±0.0072 | 0.0571±0.0140 |

| COMPAS | MLP | | |
|---|---|---|---|
|  | Acc. | $\Delta DP$ | $\Delta EOd$ |
| Fair-Mixup | 0.6620±0.0036 | 0.0562±0.0155 | 0.0735±0.0214 |
| FAAP | 0.6751±0.0039 | 0.1246±0.0196 | 0.1925±0.0391 |
| FDR | 0.6690±0.0007 | 0.0363±0.0070 | 0.0231±0.0088 |
| TabFairGAN | 0.6701±0.0094 | 0.0603±0.0224 | 0.0759±0.0397 |
| Fair-CDA | 0.6622±0.0025 | 0.2372±0.0140 | 0.4468±0.0310 |
| Ours | 0.6701±0.0009 | 0.0173±0.0079 | 0.0276±0.0189 |

| German | Logistic Regresesion | | |
|---|---|---|---|
| | Acc. | $\Delta DP$ | $\Delta EOd$ |
| Fair-Mixup | 0.7530±0.0087 | 0.0669±0.0100 | 0.2145±0.0333 |
| FAAP | 0.7205±0.0266 | 0.1006±0.0642 | 0.3152±0.1086 |
| FDR | 0.7450±0.0160 | 0.0206±0.0210 | 0.1443±0.0404 |
| TabFairGAN | 0.7445±0.0220 | 0.0624±0.0339 | 0.2511±0.0622 |
| Fair-CDA | 0.7885±0.0071 | 0.0609±0.0199 | 0.2952±0.0612 |
| Ours | 0.7955±0.0076 | 0.0103±0.0077 | 0.1252±0.0174 |
| German | MLP | | |
| | Acc. | $\Delta DP$ | $\Delta EOd$ |
| Fair-Mixup | 0.7615±0.0180 | 0.1137±0.0382 | 0.1644±0.0431 |
| FAAP | 0.7855±0.0101 | 0.0379±0.0235 | 0.2251±0.0814 |
| FDR | 0.7650±0.0000 | 0.0284±0.0000 | 0.2036±0.0000 |
| TabFairGAN | 0.7335±0.0136 | 0.0676±0.0317 | 0.2274±0.0878 |
| Fair-CDA | 0.7385±0.0147 | 0.0655±0.0287 | 0.2582±0.0882 |
| Ours | 0.7370±0.0060 | 0.0129±0.0006 | 0.1766±0.0021 |

| Drug | Logistic Regresesion | | |
|---|---|---|---|
| | Acc. | $\Delta DP$ | $\Delta EOd$ |
| Fair-Mixup | 0.6634±0.0096 | 0.1670±0.0458 | 0.2728±0.0403 |
| FAAP | 0.6605±0.0134 | 0.3225±0.0788 | 0.5616±0.1664 |
| FDR | 0.6714±0.0068 | 0.2328±0.0110 | 0.3726±0.0234 |
| TabFairGAN | 0.5772±0.0903 | 0.2125±0.1002 | 0.4187±0.1179 |
| Fair-CDA | 0.6679±0.0081 | 0.3288±0.0366 | 0.5707±0.0769 |
| Ours | 0.6607±0.0870 | 0.0812±0.0209 | 0.1160±0.0410 |
| Drug | MLP | | |
| | Acc. | $\Delta DP$ | $\Delta EOd$ |
| Fair-Mixup | 0.6310±0.0179 | 0.0265±0.0235 | 0.1336±0.0450 |
| FAAP | 0.6682±0.0107 | 0.2467±0.0137 | 0.4049±0.0284 |
| FDR | 0.6485±0.0040 | 0.2310±0.0016 | 0.4007±0.0053 |
| TabFairGAN | 0.6316±0.0203 | 0.1704±0.0373 | 0.2708±0.0695 |
| Fair-CDA | 0.6753±0.0060 | 0.3534±0.0216 | 0.6227±0.0419 |
| Ours | 0.6416±0.0073 | 0.0603±0.0286 | 0.0655±0.0451 |

## H  DATASET DETAILS

**UCI Adult Dataset.** Adult dataset (Dua et al., 2017) contains 48,842 individuals' information about income obtained from the 1994 US Census database. The target label is binarized to determine whether the income exceeds $50K/yr. Similar to (Mroueh et al., 2021) and (Yurochkin et al., 2019), samples including missing values are dropped so that the number of available samples is 45,222. The sex feature is used as a sensitive attribute.

| CelebA Smile | ResNet-18 | | | CelebA Hair | ResNet-18 | | |
|---|---|---|---|---|---|---|---|
| | Acc. | $\Delta DP$ | $\Delta EOd$ | | Acc. | $\Delta DP$ | $\Delta EOd$ |
| Fair-Mixup | 0.9092 | 0.1473 | 0.0575 | Fair-Mixup | 0.7887 | 0.3849 | 0.6369 |
| FAAP | 0.9270 | 0.1721 | 0.0735 | FAAP | 0.8180 | 0.3551 | 0.5702 |
| FDR | 0.9214 | 0.1537 | 0.0385 | FDR | 0.8181 | 0.3691 | 0.4149 |
| Ours | 0.9213 | 0.1529 | 0.0377 | Ours | 0.8230 | 0.3477 | 0.4080 |

**COMPAS Dataset.** COMPAS dataset (Jeff Larson & Angwin, 2016) contains 7,214 samples about criminal defendants and risk of recidivism with 8 attributes. It aims to classify whether a person commits a crime in the two years after they were scored. The sex feature is used as a sensitive attribute.

**German Credit Dataset.** German dataset (Dua et al., 2017) contains the credit profiles for 1,000 individuals with 20 attributes such as accounts, income, properties, and gender. The prediction goal is to classify whether a person has good or bad credit risks. The gender feature is used as a sensitive attribute.

**Drug Consumption Dataset.** Drug Consumption dataset (Dua et al., 2017) contains records from 1,885 respondents about drug consumption. Each data point has 12 attributes including the level of education, age, gender, and so on. The original task is multi-classification for 7 classes of whether and when respondents experienced drugs, but our prediction goal is abridged whether they consumed cocaine or not. The gender feature is used as a sensitive attribute.

**CelebA Dataset.** CelebA dataset (Liu et al., 2018) contains more than 200,000 celebrity face images, each coupled with 40 human-annotated binary characteristics such as gender. From these characteristics, we specifically choose smile and wavy hair, utilizing them to establish three binary classification assignments, with gender regarded as the sensitive attribute. We select these particular attributes as, in every task, a sensitive group is present which has a higher number of positive samples compared to the other.

Table 3: Features used from the Adult, COMPAS, German Credit, and Drug Consumption datasets.

| **Adult** | | | |
|---|---|---|---|
| age | workclass | education-num | marital-status |
| occupation | relationship | race | sex |
| capital-gain | capital-loss | hours-per-week | |

| **COMPAS** | | | |
|---|---|---|---|
| sex | age_cat | race | juv_fel_count |
| juv_misd_count | juv_other_count | priors_count | c_charge_degree |

| **German** | | | |
|---|---|---|---|
| Checking Account | Duration | Credit history | Purpose |
| Credit amount | Savings account | Employment | Installment rate |
| Gender | Debtors/guarantors | Residence | Property |
| Age | Installment plans | Housing | Existing credits |
| Job | Liability | Telephone | Foreigner credits |

| **Drug** | | | |
|---|---|---|---|
| Age | Gender | Education | Country |
| Ethnicity | Nscore | Escore | Oscore |
| Ascore | Cscore | Impulsive | SS |

# I ADDITIONAL EXPERIMENTS

## I.1 MULTI-LABEL CLASSIFICATION SCENARIO

We clarify that ALFA can be applied to the multi-label classification with binary-protected features as it can be seen in multiple binary classification scenarios having individual decision boundaries. In this case, the fairness loss is newly defined as covariance between a sensitive attribute and the mean of the signed distances, $L_{fair} = Cov(a, \frac{1}{T} \sum_{t=1}^{T} g_t(z_t + \delta_t))$ where $T$ is the number of targeted prediction.

Luckily, one of our datasets, the Drug Consumption dataset (Dua et al., 2017) has multiple labels. To further investigate the feasibility of our framework for the multi-label classification, we conduct additional experiments on the Drug Consumption dataset choosing four prediction goals, Cocaine, Benzodiazepine, Ketamine, and Magic Mushrooms while only Cocaine is considered as a prediction goal in the manuscript. The experimental result shows that ALFA effectively mitigates biases in the multi-label classification.

| Accuracy | Cocaine | Benzos | Ketamine | Mushrooms |
|---|---|---|---|---|
| Logistic Regression | $0.7057 \pm 0.0099$ | $0.6689 \pm 0.0113$ | $0.6989 \pm 0.0267$ | $0.7223 \pm 0.0094$ |
| Logistic Regression + ALFA | $0.6816 \pm 0.0114$ | $0.6643 \pm 0.0122$ | $0.7505 \pm 0.0023$ | $0.7307 \pm 0.0082$ |
| MLP | $0.6802 \pm 0.0144$ | $0.6527 \pm 0.0138$ | $0.7551 \pm 0.0094$ | $0.7053 \pm 0.0114$ |
| MLP + ALFA | $0.6701 \pm 0.0057$ | $0.6138 \pm 0.0036$ | $0.7343 \pm 0.0031$ | $0.6587 \pm 0.0057$ |
| $\Delta DP$ | Cocaine | Benzos | Ketamine | Mushrooms |
| Logistic Regression | $0.2691 \pm 0.0232$ | $0.3597 \pm 0.0298$ | $0.2478 \pm 0.1140$ | $0.4151 \pm 0.0372$ |
| Logistic Regression + ALFA | $0.0986 \pm 0.0289$ | $0.2666 \pm 0.0424$ | $0.0248 \pm 0.0070$ | $0.3993 \pm 0.0425$ |
| MLP | $0.2183 \pm 0.0222$ | $0.3179 \pm 0.0278$ | $0.0903 \pm 0.1320$ | $0.4072 \pm 0.0206$ |
| MLP + ALFA | $0.0760 \pm 0.0114$ | $0.1808 \pm 0.0137$ | $0.0368 \pm 0.0103$ | $0.2384 \pm 0.0099$ |
| $\Delta EOd$ | Cocaine | Benzos | Ketamine | Mushrooms |
| Logistic Regression | $0.4411 \pm 0.0483$ | $0.6448 \pm 0.0635$ | $0.5184 \pm 0.2320$ | $0.7096 \pm 0.0732$ |
| Logistic Regression + ALFA | $0.1234 \pm 0.0471$ | $0.4498 \pm 0.0858$ | $0.0689 \pm 0.0158$ | $0.6621 \pm 0.0911$ |
| MLP | $0.3505 \pm 0.0449$ | $0.5601 \pm 0.0597$ | $0.2492 \pm 0.0385$ | $0.6912 \pm 0.0441$ |
| MLP + ALFA | $0.0963 \pm 0.0249$ | $0.2971 \pm 0.0193$ | $0.1215 \pm 0.0153$ | $0.3628 \pm 0.0185$ |

Table 4: Experimental results for multi-label classification

## I.2 MULTIPLE SENSITIVE ATTRIBUTE SCENARIO

In the binary classification with multi-protected features, the Differential Fairness (DF) is measured by binarization of each multi-protected features. For example, Foulds et al. (2020) defined DF

$$DF = \max_{i,j \in S} \Big( \max(|\log \frac{P(y = 1|a = i)}{P(y = 1|a = j)}|, |\log \frac{P(y = 1|a = i)}{P(y = 1|a = j)}|\Big)$$

where $i, j \in S$, and $S$ denotes the set of the multiple sensitive attributes. Therefore, in the multi-protected feature case, we can define 'unfair region' by finding a particular sensitive attribute provoking the maximum mistreatment and reducing the misclassification rate of the unfair region as well as the binary sensitive attribute case.

For the multiple sensitive attribute setting, we adopt COMPAS dataset and MEPS dataset. MEPS (Bellamy et al., 2018) data consists of 34,655 instances with 41 features(e.g. demographic information, health services records, costs, etc.) Among all the features, only 42 features are used. The sum of total medicare visiting is used as a binary target label. When the total number of visiting is greater or equal to 10, a patient is labeled as 1, otherwise 0. And 'race' is used as multiple sensitive attributes, 0 for White, 1 for Black, and 2 for others. The experimental result shows that ALFA is also applicable to the multiple sensitive attributes scenario.

| COMPAS | Acc. | DF |
|---|---|---|
| MLP | 0.6875±0.0048 | 1.7500±0.5794 |
| MLP + ALFA | 0.6895±0.0023 | 1.3960±0.0892 |
| MEPS | Acc. | DF |
| MLP | 0.6208±0.0137 | 0.2900±0.0700 |
| MLP + ALFA | 0.6860±0.0024 | 0.1985±0.0226 |

Table 5: Experimental results for multiple sensitive attributes fairness

### I.3 MULTI-CLASS CLASSIFICATION SCENARIO

For the multi-class classification, the decision boundaries are not linear, so our framework might not be directly applicable. However, multi-class classification can indeed be conceptualized as multiple binary classifications in a certain strategy called **One-Vs-All**. In this approach, for a problem with $N$ classes, we can create $N$ different binary classifiers. Each classifier is trained to distinguish between one of the classes and all other classes combined.

As each classifier can be seen as a binary classification task, we can utilize ALFA for the multi-class classification scenario by detecting unfair regions and recovering the region by fairness attack. The evaluation metric for multi-class fairness takes maximum Demographic Parity across the classes (Denis et al., 2021). In details,

$$\Delta DP_{\text{multi}} = \max_{k \in [K]} \left| P(\hat{Y} = k | a = 1) - P(\hat{Y} = k | a = 0) \right|$$

where $\hat{Y}$ is the predicted class, and $k \in [K]$ denotes each class $k$ in the multi-class classification.

Among existing datasets for fairness research, Drug dataset can be used for multi-class classification. In fact, the original labels of the Drug dataset are multi-class settings, from 'CL0' to 'CL6' indicating the frequency of drug abuse. We have binarized them as 'never used' and 'ever used' regardless of the frequency in the main paper. However, for the multi-class classification setting, we adopt the original multi-class setting and report the mean accuracy and $\Delta DP_{\text{multi}}$ with MLP.

| Drug Multi-class | Acc. | $\Delta DP_{\text{multi}}$ |
|---|---|---|
| MLP | 0.5196±0.0032 | 0.1930±0.0132 |
| MLP + ALFA | 0.4960±0.0219 | 0.1733±0.0287 |

Table 6: Experimental results for multi-class classification

### I.4 MORE COMPLEX MODEL

To verify the effectiveness of our framework on the tabular dataset, we extend the experiment using 1) a ResNet-like backbone, and 2) a Transformer-like backbone designed for tabular datasets both proposed in (Gorishniy et al., 2021). The figures of Pareto Frontier show that ALFA is applicable to more complex model.

### I.5 MORE COMPLEX DATASET (NLP)

Moreover, we further explore the adaptability of the proposed method to the Natural Language Processing (NLP) dataset. We utilize Wikipedia Talk Toxicity Prediction (Thain et al., 2017) which is a comprehensive collection aimed at identifying toxic content within discussion comments posted on Wikipedia's talk pages, produced by the Conversation AI project. In this context, toxicity is defined as content that may be perceived as "rude, disrespectful, or unreasonable." It consists of over 100,000 comments from the English Wikipedia, each meticulously annotated by crowd workers, as delineated in their associated research paper. A challenge presented by this dataset is the

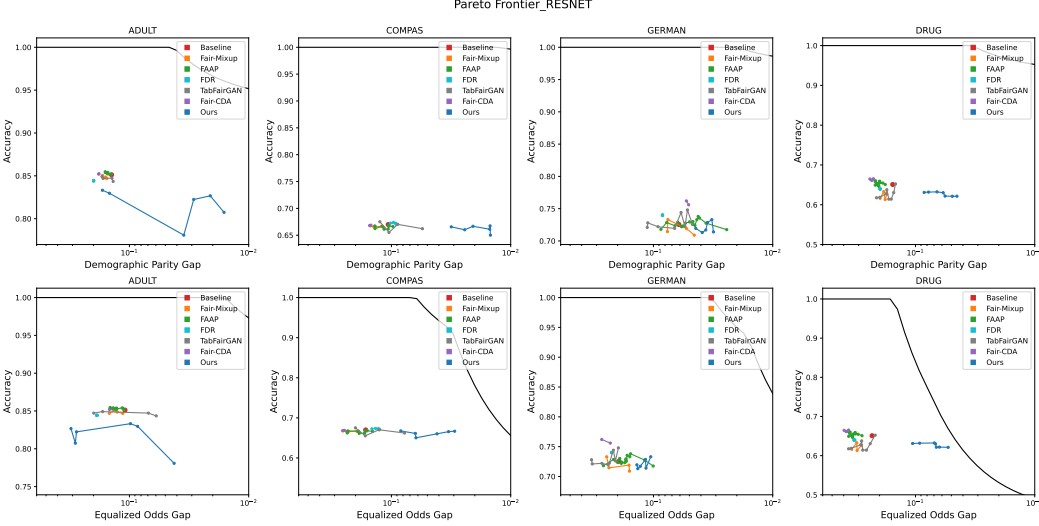

Figure 6: Pareto Frontier for ResNet (Gorishniy et al., 2021).

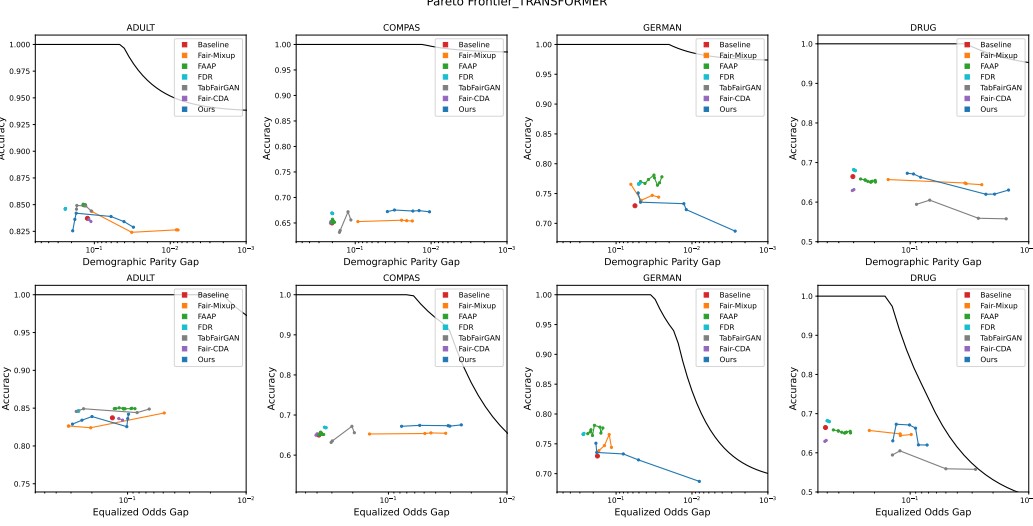

Figure 7: Pareto Frontier for FT-Transformer (Gorishniy et al., 2021).

underrepresentation of comments addressing sensitive subjects such as sexuality, religion, gender identity, and race. In this paper, the existence of sexuality terms such as 'gay', 'lesbian', 'bisexual', 'homosexual', 'straight', and 'heterosexual' is used as the sensitive attribute, 1 for existing, and 0 for absence.

For the NLP dataset, we use a pre-trained word embedding model, Glove (Pennington et al., 2014) to produce input data and use MLP and LSTM network (Hochreiter & Schmidhuber, 1997) for the experiments.

## J    ANOTHER FAIRNESS CONSTRAINT

In this part, we show that ALFA can adopt any types of fairness constraint during the fairenss attack. As an alternative of (Zafar et al., 2017), we present (Wu et al., 2019) below.

| Toxicty | Accuracy | $\Delta DP$ | $\Delta EOd$ |
|---|---|---|---|
| MLP | $0.9345 \pm 0.0043$ | $0.1735 \pm 0.0066$ | $0.0734 \pm 0.0145$ |
| MLP + ALFA | $0.9349 \pm 0.0002$ | $0.1684 \pm 0.0019$ | $0.0500 \pm 0.0049$ |
| LSTM | $0.9294 \pm 0.0007$ | $0.1922 \pm 0.0053$ | $0.0806 \pm 0.0095$ |
| LSTM + ALFA | $0.9280 \pm 0.1001$ | $0.1872 \pm 0.0001$ | $0.0607 \pm 0.0004$ |

Table 7: Experimental results for NLP dataset

Let's say $f(\mathbf{X})$ is a logit of binary classifier given data $\mathbf{X}$ and define indicator functions $\mathbb{1}(\cdot)$ where $\cdot$ denotes each condition for the indicator function.

The empirical DP Gap is

$$\Delta DP(f) = \frac{1}{|\mathbb{1}(a=1)|} \sum_{a=1} \mathbb{1}(f(\mathbf{X}) > 0) - \frac{1}{|\mathbb{1}(a=0)|} \sum_{a=0} \mathbb{1}(f(\mathbf{X}) > 0).$$

and can be rewritten in the expected form as

$$\Delta DP(f) = \mathbb{E}\big[ \frac{\mathbb{1}(a=1)}{p_1} \mathbb{1}(f(\mathbf{X}) > 0) - (1 - \frac{\mathbb{1}(a=0)}{1-p_1} \mathbb{1}(f(\mathbf{X}) < 0)) \big]$$

where $p_1 = p(a=1)$.

Moreover, the relaxed form replacing the indicator function to real-valued function is written as

$$\Delta DP(f) = \mathbb{E}\big[ \frac{\mathbb{1}(a=1)}{p_1} f(\mathbf{X}) - (1 - \frac{\mathbb{1}(a=0)}{1-p_1} f(\mathbf{X})) \big].$$

In (Wu et al., 2019), $f(\mathbf{X})$ is replaced again to construct a convex form using two different surrogate functions to use $\Delta DP$ as a fairness constraint,

$$\Delta DP\kappa(f) = \mathbb{E}\big[ \frac{\mathbb{1}(a=1)}{p_1} \kappa(f(\mathbf{X})) - \big(1 - \frac{\mathbb{1}(a=0)}{1-p_1} \kappa(-f(\mathbf{X})) \big) \big]$$

$$\Delta DP\delta(f) = \mathbb{E}\big[ \frac{\mathbb{1}(a=1)}{p_1} \delta(f(\mathbf{X})) - \big(1 - \frac{\mathbb{1}(a=0)}{1-p_1} \delta(-f(\mathbf{X})) \big) \big]$$

where $\kappa$ is a convex surrogate function $\kappa(z) = \max(z+1, 0)$ and $\delta$ is a concave surrogate function $\delta(z) = \min(z, 1)$ as proposed in (Wu et al., 2019). If $\Delta DP(f) \geq 0$, we directly use $\Delta DP\kappa(f)$ as a fairness constraint, otherwise use $\Delta DP\delta(f)$,

$$L_{fair} = \begin{cases} \Delta DP\kappa(f) & \text{if } \Delta DP \geq 0 \\ \\ \Delta DP\delta(f) & \text{if } \Delta DP < 0. \end{cases}$$

Also, it can be extended to use $\Delta EOD$ directly as a fairness constraint, by conditioning $\Delta DP$ for each $y \in \{0, 1\}$.

$$\Delta EOD = \Big[ \frac{1}{|\mathbb{1}(a=1, y=1)|} \sum_{a=1, y=1} \mathbb{1}(f(x) > 0) - \frac{1}{|\mathbb{1}(a=0, y=1)|} \sum_{a=0, y=1} \mathbb{1}(f(x) > 0) \Big]$$

$$+ \Big[ \frac{1}{|\mathbb{1}(a=1, y=0)|} \sum_{a=1, y=0} \mathbb{1}(f(x) > 0) - \frac{1}{|\mathbb{1}(a=0, y=0)|} \sum_{a=0, y=0} \mathbb{1}(f(x) > 0) \Big],$$

and can be rewritten in the expected form as

$$\Delta EOD(f) = \mathbb{E}\big[ \frac{\mathbb{1}(a=1, y=1)}{p_{1,1}} \mathbb{1}(f(\mathbf{X}) > 0) - \big(1 - \frac{\mathbb{1}(a=0, y=1)}{\pi - p_{1,1}} \mathbb{1}(f(\mathbf{X}) < 0) \big) \big]$$

$$+\mathbb{E}\Big[\frac{\mathbb{1}(a=1,y=0)}{p_{1,0}}\mathbb{1}(f(\mathbf{X})>0)-\big(1-\frac{\mathbb{1}(a=0,y=0)}{1-\pi-p_{1,0}}\mathbb{1}(f(\mathbf{X})<0))\big)\Big]$$

since $1=\mathbb{E}[\frac{\mathbb{1}(a=0,y=1)}{p_{0,1}}]=\mathbb{E}[\frac{\mathbb{1}(a=0,y=1)}{\pi-p_{1,1}}]=\mathbb{E}[\frac{\mathbb{1}(a=0,y=1)}{\pi-p_{1,1}}\mathbb{1}(f(\mathbf{X})<0)+\frac{\mathbb{1}(a=0,y=1)}{\pi-p_{1,1}}\mathbb{1}(f(\mathbf{X})>0)]$ and $1=\mathbb{E}[\frac{\mathbb{1}(a=0,y=0)}{p_{0,0}}]=\mathbb{E}[\frac{\mathbb{1}(a=0,y=0)}{1-\pi-p_{1,0}}]=\mathbb{E}[\frac{\mathbb{1}(a=0,y=0)}{1-\pi-p_{1,0}}\mathbb{1}(f(\mathbf{X})<0)+\frac{\mathbb{1}(a=0,y=0)}{1-\pi-p_{1,0}}\mathbb{1}(f(\mathbf{X})>0)]$, $\pi=p(y=1)$ and $p(y=0)=1-\pi$ where $p_{1,1}=P(a=1,y=1)$ and $p_{1,0}=P(a=1,y=0)$. $\Delta EOD$ can be expressed as a convex form,

$$\Delta EOD\kappa(f)=\mathbb{E}\Big[\frac{\mathbb{1}(a=1,y=1)}{p_{1,1}}\kappa(f(\mathbf{X}))-\big(1-\frac{\mathbb{1}(a=0,y=1)}{\pi-p_{1,1}}\kappa(-f(\mathbf{X})))\big)\Big]$$
$$+\mathbb{E}\Big[\frac{\mathbb{1}(a=1,y=0)}{p_{1,0}}\kappa(f(\mathbf{X}))-\big(1-\frac{\mathbb{1}(a=0,y=0)}{1-\pi-p_{1,0}}\kappa(-f(\mathbf{X})))\big)\Big]$$

$$\Delta EOD\delta(f)=\mathbb{E}\Big[\frac{\mathbb{1}(a=1,y=1)}{p_{1,1}}\delta(f(\mathbf{X}))-\big(1-\frac{\mathbb{1}(a=0,y=1)}{\pi-p_{1,1}}\delta(-f(\mathbf{X})))\big)\Big]$$
$$+\mathbb{E}\Big[\frac{\mathbb{1}(a=1,y=0)}{p_{1,0}}\delta(f(\mathbf{X}))-\big(1-\frac{\mathbb{1}(a=0,y=0)}{1-\pi-p_{1,0}}\delta(-f(\mathbf{X})))\big)\Big].$$

where

$$L_{fair}=\begin{cases}\Delta EOD\kappa(f) & \text{if } \Delta EOD\geq 0\\ \\ \Delta EOD\delta(f) & \text{if } \Delta EOD<0.\end{cases}$$

Therefore, different from the covariance (Zafar et al., 2017) between prediction and sensitive attribute, the convex fairness constraint takes into account the empirical outputs considering all potential dependencies, not focusing on a particular attribute.

## K  ANALYSIS ON SYNTHETIC DATA

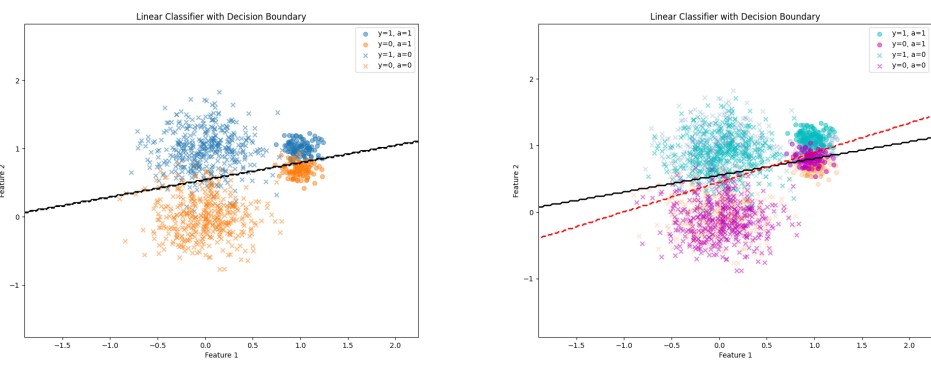

Figure 8: Decision boundary before (left) and after (right) the fairness attack. The blue and orange points indicate the original data distribution. The cyan and magenta points indicate the perturbed data from the original distribution. The decision boundary is rotated from **black** to **red** line.

## L  CAUSAL STRUCTURE

## M  THE EFFECT OF $\lambda$

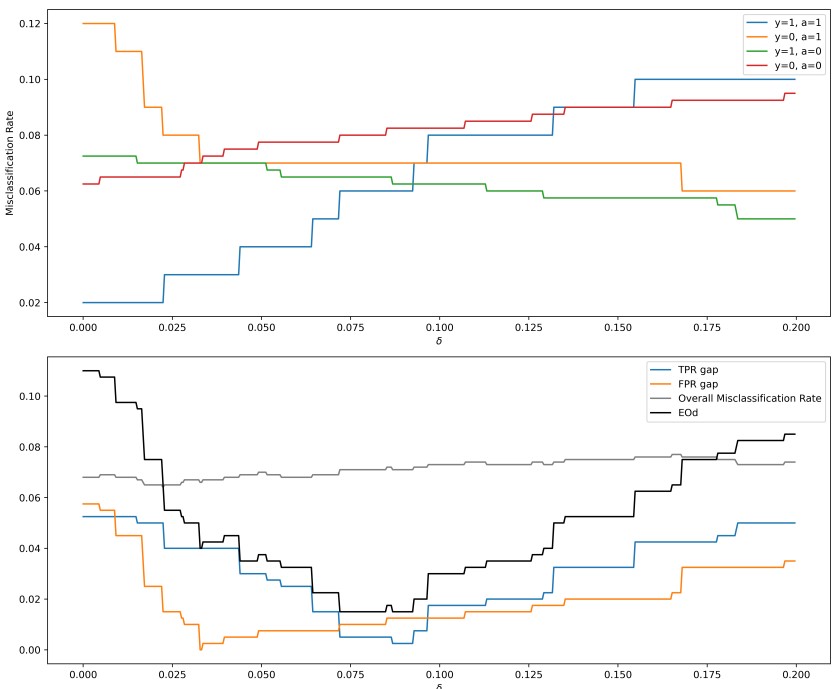

Figure 9: The impact of perturbation magnitude $\delta$ on the misclassification rate of each subgroup (upper) and the EOd (FPR gap + TPR gap). We can show that there exists a $\delta$ such that improves fairness while maintaining accuracy.

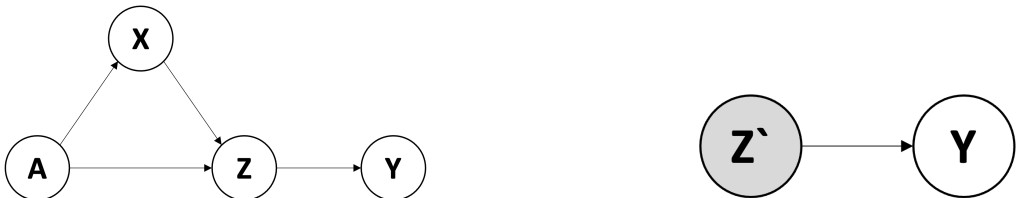

Figure 10: Structure assumption before (left) and after (right) the debiasing.

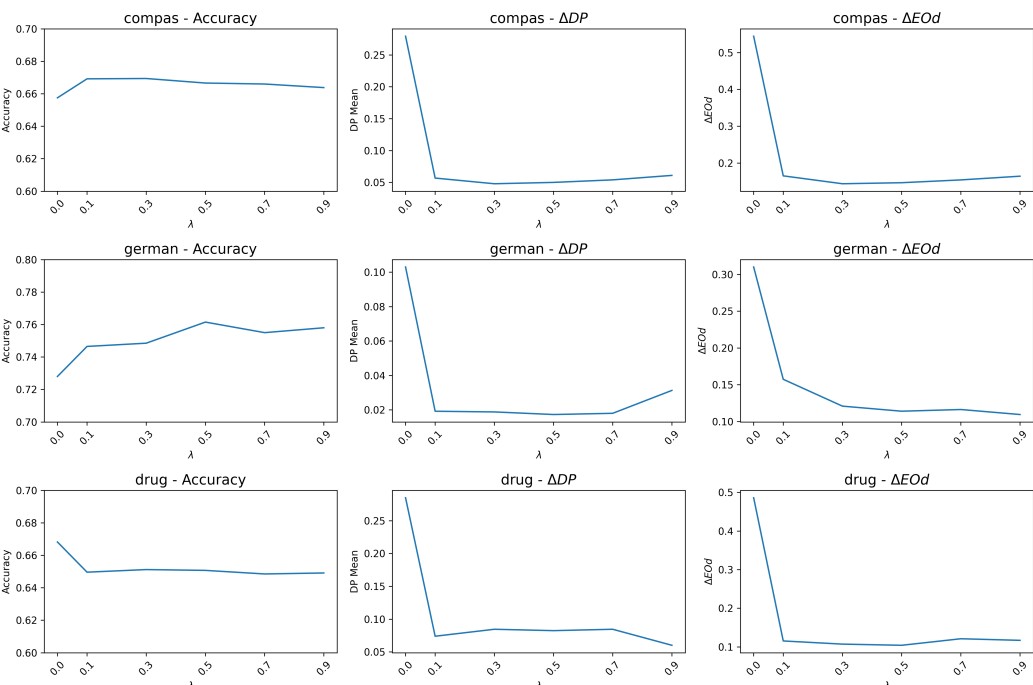

Figure 11: The effect of $\lambda$ on three datasets with Logistic Regression. Compared to the baseline($\lambda = 0$), any $\lambda$ improve fairness.

