# OpenReview forum: "Adversarial Latent Feature Augmentation for Fairness"
_ICLR.cc/2024/Conference — Submitted to ICLR 2024_

### Official Review · Reviewer_CWiz · 2023-11-01

**Soundness:** 3 good
**Presentation:** 3 good
**Contribution:** 2 fair
**Rating:** 6
**Confidence:** 3

**Summary:**

The paper proposes the Adversarial Latent Feature Augmentation (ALFA) which merges adversarial attacks into data augmentation in the latent space to promote fairness. The paper points out that adversarial perturbation can in fact be used to augment fairness. Through covariance-based fairness constraint, this paper unveils a counter-intuitive relationship between adversarial attacks against fairness and enhanced model fairness upon training with the resultant perturbed latent features by hyperplane rotation. The paper theoretically proves that the proposed adversarial fairness objective assuredly generates biased feature perturbation, and empirically validate that training with adversarial features significantly improve fairness.

**Strengths:**

This paper has good originality, high quality and clear expression.  The paper combines adversarial attacks into data augmentation in the latent space to promote fairness and proposes a new method.

**Weaknesses:**

More complex dataset is better to be verified the effectiveness of the proposed method.

**Questions:**

1.Adversarial perturbations suffer from robust overfitting[1] when used as data augmentation in adversarial training, when used as data augmentation in fairness, is there exits some overfitting in promoting fairness?
2.The rotation of the decision boundary is because of the exists of adversarial pertuabations in the proposed method,will different forms of perturbations such as L2 perturbation or L-infinity effect the rotation of decision boundary?
3.The proposed method has a relatively large number of hyperparameters, will this affect its efficiency when applied to more complex datasets?



[1]Leslie Rice, Eric Wong, and Zico Kolter. Overfitting in adversarially robust deep learning. In International Conference on Machine Learning, pp. 8093–8104. PMLR, 2020.

---

> ### Author Response · Authors · 2023-11-18
>
> ### More complex dataset
> Please see the global response, addressing more general scenarios beyond binary classification with binary-sensitive attributes.
>
> ### Overfitting problem in data augmentation with adversarial training
> In the proposed framework, overfitting is not a matter, because unlikely typical adversarial training, we use different objective functions in the attacking step and training step. Moreover, we can interpret the 'unfair region' described in Figure 1(a) as an overfitted or underfitted area. We directly recover the unfair region making the misclassification rates for each subgroup fair, and alleviate the overfitting and underfitting of original prediction.
>
> ### About the L2 perturbation and hyperparameters
> In fact, our framework is not limited to $L_\infty$ perturbation. $L2$ perturbation is also applicable because the perturbation vector $\delta$ is trained by the fairness attack objective function Eq.(5). Especially, the Sinkhorn distance regularizes the perturbated dataset has a similar distribution with the original dataset.
>
> The reason we adapt $L_\infty$ is the hyperparameter $\epsilon$ is not straightforward if we use $L2$ perturbation. However, we may be able to omit the hyperparameter $\epsilon$ by computing the proper $\epsilon$ given dataset and decision boundary. In detail, we can measure the Euclidean distance between each subgroup and the pre-traind decision boundary. And we can set the mean distance as $\epsilon$ for the $L2$ perturbation.
>
> For example, if we use $L2$ perturbation by setting $\epsilon$ as the measured distance for COMPAS dataset and MLP with $\alpha=1.0$ and $\lambda = 0.5$, we obtain the range of $\delta\in[-0.2722,0.2552]$. Following the measured perturbation, we obtain a similar result with $L_\infty$ by setting $\epsilon=0.27$. Therefore, we can conclude that the choice of $L2$ and $L_\infty$ doesn't affect the performance. If a practitioner wants to control the hyperparameter $\epsilon$, $L_\infty$ is suitable. On the other hand, if one wants to simplify the setting, the $\epsilon$ measurement with $L2$ perturbation could be a wise choice. We will add this ablation study in the revision.
>
>
> | COMPAS | magnitude of $\delta$| |Accuracy |  | $\Delta DP$ |  | $\Delta EOd$ |  |
> | --- | --- | --- | --- | --- | --- | --- |--- | --- |
> | Method |min $\delta$ | max $\delta$ | mean | std. | mean | std. | mean | std. |
> | $L2$-perturbation |-0.2722|0.2552| 0.6664 | 0.0029 | 0.0108 | 0.0056 | 0.0663 | 0.0215 |
> | $L_\infty$-perturbation |-0.2700|0.2700| 0.6613| 0.0033| 0.0126| 0.0065| 0.0686| 0.0212|
>
> When it comes to $\lambda$, the selection of $\lambda$ is significant to the accuracy-fairness trade-off since it works as the weight between the original and perturbed dataset. Therefore, we vary $\lambda$ and report the 'line' of the performances in the Pareto Frontier as well as varying $\alpha$ in Eq.(5). We suggest using $\lambda=0.5$ since it empirically shows the best accuracy-fairness trade-off for all datasets. Still, the hyperparameter tuning is needed for $\alpha$. Here we report the training time for experiments for each combination of hyperparameters. Each fine-tuning is conducted in 50 epochs. We measure the training time for a single run for baseline, and the average of 10 runs for fine-tuning.
>
> | Dataset | Baseline | Fine-tuning|
> | --- | --- | --- |
> | Adult | 40.046s| 37.018s |
> | COMPAS |9.118s| 6.166s|
> | German |3.934s |1.177s|
> | Drug| 4.748s | 1.838s|
>
> Therefore, we can omit the hyperparameter $\epsilon$ and $\lambda$ by using $L2$ perturbation taking the mean Euclidean distance between the latent feature and the decision boundary as $\epsilon$ and setting $\lambda=0.5$ to make the original dataset and perturbed dataset equally contribute. However, even though only one hyperparameter, $\alpha$, is need to be tuned, the hyperparameter tunig itself takes less time compared to the original ERM because fine-tuning trains only the last layer of the model.

---

> > ### Comment · Reviewer_CWiz · 2023-11-22
> >
> > Thanks to the authors for their reply, I have no more questions.

---

### Official Review · Reviewer_XHAN · 2023-11-01

**Soundness:** 2 fair
**Presentation:** 3 good
**Contribution:** 2 fair
**Rating:** 3
**Confidence:** 1

**Summary:**

The paper discusses a novel approach called Adversarial Latent Feature Augmentation (ALFA) to address bias in machine learning models. This approach involves perturbing latent features with the aim of mitigating bias and achieving fairness. The authors define a fairness attack as a feature-level perturbation to maximize a covariance-based fairness constraint, which can lead to biased latent features. They argue that fine-tuning these perturbed features can rotate the decision boundary, covering unfair regions and achieving group fairness. The method is tested on various datasets, including Adult, COMPAS, German, and Drug datasets, as well as image classification on CelebA, demonstrating accuracy while achieving group fairness.

**Strengths:**

1. The paper is clearly presented and easy to follow.
2. It provides some theory analysis.
3. Empirical studies sound.

**Weaknesses:**

1. Why "GAN-based perturbation might yield unsuitable generated features not aligning on the sensitive hyperplane" for tabular datasets? The intuition is unclear to me. It would be better to explain using concrete examples and provide citations.
2. Can the proposed method be applied to non-tabular datasets, such as vision datasets (i.e., images and videos)? If not, the scope of the method is limited. If so, why does W1 matter?
3. In Eq.(1), the perturbation \delta should be bolded. What is the \bar{a} in Eq.(1)? Is this the mean of all a_i? The notation of \bar{a} is not defined before using.
4. Adversal fairness learning is not novel. The earliest work dates back to 2018 [1]. I doubt the novelty of the proposed method in 3.3.
5. According to the setting and empirical studies, the proposed method can be applied only when there is only one sensitive attribute with binary values.


[1] Christina Wadsworth, Francesca Vera, Chris Piech. Achieving Fairness through Adversarial Learning: an Application to
Recidivism Prediction. FAT/ML Workshop, 2018

**Questions:**

See weaknesses.

---

> ### Author Response · Authors · 2023-11-18
>
> ### The statement about GAN in tabular dataset
> The statement in the paper reviewer mentioned is based on our empirical observation of the existing work, FAAP [1] which is applicable only to the image dataset. As shown in Figures 1 and 2 in the paper demonstrating the experimental result on tabular datasets, FAAP shows poor performance in tabular datasets and we suspect it is because the GAN-based perturbation might not be converged well.
>
>
> ### Application to vision dataset
> We reported the experimental results for the image dataset, CelebA as shown in Figure 4. In this case, FAAP shows comparable results to ours, and W1 is not a matter in this case. Therefore, it shows that FAAP works only for image datasets, while ALFA (ours) works for both tabular and image datasets. Moreover, we conduct additional experiments applying ALFA to the NLP dataset, as shown in the global response in this rebuttal.
>
> ### Missing notations
> Thanks for pointing out the typos. We will revise them and screen again the entire notations thoroughly.
>
>
> ### The novelty of adversarial fairness
> ALFA and [2] have different perspectives on adversarial training. [2] uses an adversary to predict sensitive attributes, while the classifier predicts the label. The objective function for the classifier is designed to minimize the binary cross-entropy (BCE) loss on the label while maximizing BCE loss on the sensitive attribute, which makes the classifier not use the demographic information during the prediction. However, no mathematical derivation is introduced as to how maximizing BCE on the sensitive attribute makes the classifier fair. Moreover, [2] is designed only for the COMPAS dataset.
>
> We contain recent adversarial training methods in the paper such as [1] and [3], and describe how they are different from ours, in Section 1 and Section 2, respectively.
>
>
> ### Beyond binary labels and sensitive attributes
> Please see the global response, addressing more general scenarios beyond binary classification with binary-sensitive attributes.
>
>
>
>
> [1] Zhibo Wang, Xiaowei Dong, Henry Xue, Zhifei Zhang, Weifeng Chiu, Tao Wei, and Kui Ren. Fairness-aware adversarial perturbation towards bias mitigation for deployed deep models. In Proceedings of the IEEE/CVF Conference on Computer Vision and Pattern Recognition, pp. 10379-10388, 2022b
>
> [2] Christina Wadsworth, Francesca Vera, Chris Piech. Achieving Fairness through Adversarial Learning: an Application to Recidivism Prediction. FAT/ML Workshop, 2018
>
> [3] Peizhao Li, Ethan Xia, and Hongfu Liu. Learning antidote data to individual unfairness. In International Conference on Machine Learning, pp. 20168–20181. PMLR, 2023.

---

### Official Review · Reviewer_KRtS · 2023-11-01

**Soundness:** 2 fair
**Presentation:** 2 fair
**Contribution:** 3 good
**Rating:** 6
**Confidence:** 3

**Summary:**

The paper proposes using adversarial attack on fairness of a Deep network to generate more synthetic samples that could be used to fix the bias present in the model. The authors show through a small example how generated adversarial samples would be overlapping with the region of unfairness. Hence using such samples in the training set rotates the classifier boundary in a way the misclassification regions are now correctly predicted. The idea is quite novel application of adversarial attack on a neural network. The paper also provides theory on why optimizing over covariance is relevant to impacting the fairness constraints like DP and Equality of Opportunity. The experiment results on standard datasets show better pareto frontier for proposed method for most cases.

**Strengths:**

The paper provides insights on one of the reasons of unfairness due to data imbalance and tackles the problem through data augmentation. I like the analysis presented in the paper.

1. The proposed data augmentation intelligently utilizes the adversarial attack to identify the unfairness regions. The samples from unfairness region are then used to teach the model to fix its linear decision boundary.
2. The optimization objective for adversarial attack are tied to the fairness objectives under certain conditions.
3. The pareto frontier for proposed technique is better than baselines.

**Weaknesses:**

There are some minor weaknesses of the proposed approach:
1. Storage cost increase due to dataset augmentation. Not necessarily a flaw.
2. Limitation: Paper only deals with binary classification and two protected classes.
3. The current solution of adjusting rotation of the linear classifier would only work when the two classes are separated in their latent space. For more complicated cases like lesser data, higher dimensional data etc. the latent space might not be separated and intuition would not hold. However this would be future direction to look into.

**Questions:**

I find the proposed method quite interesting. However I have some clarifying questions:

Q1. One of the comparisons that could be done would be to generate samples using directly optimizing empirical EOd or DP, i.e finding adversarial examples using maximizing False positive error or False negative error. This could be done by flipping the labels for the dataset. Why choose to attack L_fair vs False positive rate?

Q2. In theorem 3.2, $N_p = 4\times \max(n_{i,j})$. How is this maintained? Does this mean some samples would be repeated before adversarial attack?

Q3. The final loss function is a convex combination of true and perturbed dataset. I wanted to understand the weight attributed to the perturbed dataset. What values of $\lambda$ are used in different cases?

Q4. How can linear decision boundary handle the case of multiple classes in protected attribute? A rotation would not be able to fix the bias in the dataset.

I will adjust my score after these concerns are addressed.

---

> ### Author Response · Authors · 2023-11-18
>
> ### Storage cost
> ALFA is effective in terms of storage or memory cost. In detail, we can think about two types of augmentation, offline augmentation (storing augmented data in the drive) and online augmentation (augmentation on the data loader, stored in the memory temporarily).
>
> For the offline augmentation, more storage cost may be required compared to the baseline ALFA. However, it requires less storage cost compared to the data augmentation on the input space. For example, for $N$ samples, the storage cost for $N$ images might be huge, while ours requires a single $N \times d$ matrix where $d$ is the dimension of the latent feature.
>
> For the online augmentation, the data augmentation on the input space may require more training time and memory space. However, as ours is a fine-tuning method using a latent feature from original images and a perturbed latent feature, we only require two $N \times d$ matrices in the memory space removing the dataloader from the input images in the memory.
>
>
> ### Beyond binary $Y$ and $A$
> Please see the global response, addressing more general scenarios beyond binary classification with binary-sensitive attributes.
>
> ### Linear separability assumption
> We agree with the reviewer's point that the linear separability assumption is needed in our framework. However, linear separability is a widespread assumption [1] even though the size of the dataset is small or the number of features is large even in the input space. Moreover, deep neural networks transform the input data into well-separated latent features, making the classification easier.
>
> [1] Moshkovitz, Michal, Yao-Yuan Yang, and Kamalika Chaudhuri. "Connecting interpretability and robustness in decision trees through separation." International Conference on Machine Learning. PMLR, 2021.
>
> ### Label flipping strategy
> Although flipping labels might generate samples in which $\Delta EOd$ and $\Delta DP$ are worse, training on them doesn't necessarily improve fairness. For example, we consider the latent distribution in Figure 1 in the paper. The augmented dataset may have the same distribution as the original data but flipped labels. Training on the original and augmented (flipped) dataset together may produce poor decision boundaries, exacerbating the accuracy.
>
> On the other hand, ALFA aims to produce a biased dataset while maintaining the data distribution of the original feature, using Sinkhorn distance, which ensures the generated data has a similar data distribution to the original one and is useful to improve fairness while maintaining accuracy.
>
> ### Sample repetition
> Yes, the data sampling is repeated to maintain the size of each subgroup to be the same. Although this repetition is not a mandatory condition empirically, we make the subgroup size equal to ensure the efficiency of the fairness attack as shown in Theorem 3.2.
>
> ### Contribution of $\lambda$
> The selection of $\lambda$ is significant to the accuracy-fairness trade-off since it works as the weight between the original and perturbed dataset. Therefore, we vary $\lambda$ and report the 'line' of the performances in the Pareto Frontier in Figures 2 and 3 in the paper, as well as varying $\alpha$ in Eq.(5).
>
>
> ### Multiple protected attributes
> Please see the global response, addressing more general scenarios beyond binary classification with binary-sensitive attributes.

---

> ### Comment · Reviewer_KRtS · 2023-11-22
> **Thank you for the rebuttal**
>
> I would like to thank the authors for their rebuttal work. For most parts, my questions have been answered.
> However, I do have some additional comments on them which seems to be important -
> 1. **Linear separability**: The original comment should be read as separation of data wrt different sensitive attributes which seems to be essential to be explanation that rotation of linear boundary would help cover unfair regions. Even though data with different classes (labels) would be linearly separable [1], it is not guaranteed the same for data with different sensitive attributes. Consider Figure 1, if the two sensitive attributes were inter-mixed in latent space, I think a rotation would be insufficient to fix unfairness. This issue even worsens when there are more classes of sensitive attributes.
>
>  2. **Label flipping** - the original question Q1 was to generate augmented samples using maximizing FPR for one of the classes. This could in theory generate samples to cover the unfair region as well. How would this direct approach do in comparison to covariance loss used in the paper.
>
> 3. **Contribution of $\lambda$** - I was of the understanding that the figures 2, 3 are plots between EOd or DP vs Accuracy. Q3 was regarding the absolute value of $\lambda$ vs EOd and Accuracy.
>
> 4. **New results on multi-class classification** - The new results seem to indicate that proposed method leads to significant drop in acc with not much gain in DP. Also why are EOd not reported for this experiment?
>
> Minor typo in definition Differential Fairness DF in the rebuttal. Please check indices i,j.

---

> > ### Author Response · Authors · 2023-11-22
> >
> > Thank you for taking the time to respond to our rebuttal.
> >
> > ## Linear separability
> > If the demographic groups in the latent are perfectly inter-mixed, it indicates a fair representation in itself. Naturally, we hypothesize they have separation. In fact, literature [1,2,3] requires the linear separability of sensitive groups since they use an auxiliary classification model predicting sensitive attributes.
> > However, linear separability is not strictly required in our framework. Even if the demographic groups are partially inter-mixed, we can still define the unfair regions and the decision boundary can be rotated as long as the latent feature distribution is linearly separable in terms of labels.
> >
> > [1] Zhibo Wang, Xiaowei Dong, Henry Xue, Zhifei Zhang, Weifeng Chiu, Tao Wei, and Kui Ren. Fairness-aware adversarial perturbation towards bias mitigation for deployed deep models. In Proceedings of the IEEE/CVF Conference on Computer Vision and Pattern Recognition, pp. 10379-10388, 2022b
> >
> > [2] Christina Wadsworth, Francesca Vera, Chris Piech. Achieving Fairness through Adversarial Learning: an Application to Recidivism Prediction. FAT/ML Workshop, 2018
> >
> > [3] Ramaswamy, Vikram V., Sunnie SY Kim, and Olga Russakovsky. "Fair attribute classification through latent space de-biasing." Proceedings of the IEEE/CVF conference on computer vision and pattern recognition. 2021.
> >
> > ## Label flipping
> > Finding adversarial samples using FPR or FNR maximization looks similar to our framework, but there's a difference in terms of the direction of the perturbation. As the covariance-based fairness attack is designed to maximize the covariance between the sensitive attribute and label, the direction of perturbation is determined as perpendicular to the decision boundary to cover the unfair region effectively. However, maximizing FPR or FNR doesn’t have an obvious tendency. For example, pushing samples to outliers could be a successful FPR maximization, but it doesn’t lead to fairness improvement.
> >
> > ## Contribution of $\lambda$
> > We visualize the effect of $\lambda$ for three datasets, COMPAS, German, and Drug with Logistic Regression varying $\lambda=0.1,0.3,0.5,0.7,0.9$ and the set $\lambda=0$ as a baseline in Figure 11, Appendix M. Compared to the baseline($\lambda=0$), any $\lambda$ improve fairness. Intuitively, relying heavily on either original samples or perturbed samples might not be a good strategy. We suggest $\lambda=0.5$ as the default setting making the original and augmented features have an equal contribution.

---

> > > ### Author Response · Authors · 2023-11-22
> > >
> > > ## Experimental results for multi-class classification
> > > Here we report a new experimental result with $EOd_{\text{multi}}$. As $EOd_{\text{multi}}$ is not strictly defined in literature, we follows the definition of Predictive Equality (PE) and Equal Opportunity (EO) notion in [4] for each class, and take maximum of the summation of PE and EO similar to $DP_{\text{multi}}$. We define $EOd_{\text{multi}}$ as
> > > \begin{align}
> > > EOd_{\text{multi}} =  \max_{k\in[K]} &\Bigl( \bigl\vert P(\hat{Y}=k \vert a=1, Y=k) -P(\hat{Y}=k \vert a=0, Y=k)\bigr\vert  \\\\
> > > &+ \bigl\vert P(\hat{Y}\neq k \vert a=1, Y\neq k) -P(\hat{Y}\neq k \vert a=0, Y\neq k) \bigr \vert \Bigr).
> > > \end{align}
> > >
> > >
> > > Here we reveal the technical changes in the rebuttal period. In the binary classification, the fine-tuning learning rate of 0.01 and 0.001 didn't affect the experimental result, so we used 0.01 for convenience (including the previous result in the rebuttal). However, we found that ALFA's performance on multi-class classification is sensitive to hyperparameters including learning rate. While we fix $\lambda=0.5$, we tune other hyperparameters and obtain improved results with $\alpha=0.01$, $\epsilon=1$, and $lr=0.001$.
> > >
> > >
> > > | Drug-multi| Accuracy |  | $\Delta DP_{\text{multi}}$ |  |$\Delta EOd_{\text{multi}}$ |  |
> > > | --- | --- | --- | --- | --- |  --- | --- |
> > > | Model | mean | std. | mean | std. |mean | std. |
> > > | MLP | 0.5207|0.0024|0.1917|0.0062|0.3010|0.0158|
> > > | MLP + ALFA (lr=0.01) | 0.4851|0.0231|0.1669|0.0224|0.3185|0.0551|
> > > | MLP + ALFA (lr=0.001)| 0.5501|0.0013|0.0374|0.0323|0.1934|0.0302|
> > >
> > > We grid search the hyperpameter by fixing $\lambda=0.5$, set the searching range as $\epsilon=[0.1, 0.2,0.5,1]$, $\alpha=[0.01,0.1,1,10]$, and $lr=[0.01, 0.001]$. Each fine-tuning is conducted in 30 epochs. We measure the training time for a single run for baseline, and the average of 10 runs for fine-tuning.
> > >
> > > | Dataset | Baseline | Fine-tuning|
> > > | --- | --- | --- |
> > > | Drug (Multi)| 9.580s | 6.929s|
> > >
> > > [4] Rouzot, Julien, Julien Ferry, and Marie-José Huguet. "Learning Optimal Fair Scoring Systems for Multi-Class Classification." 2022 IEEE 34th International Conference on Tools with Artificial Intelligence (ICTAI). IEEE, 2022.
> > >
> > >
> > > ## Typo
> > > Thanks for pointing out the typo. We also found the typo regarding the notation of positive ratio and negative ratio for each log probability and changed the indices notation to cartesian product notation.
> > >
> > > $$
> > >  DF = \max_{(i,j) \in S\times S} \Bigl(\max(\bigl \vert \log \frac{P(\hat{y}=1 \mid a=i)}{P(\hat{y}=1\mid a=j)}\bigl \vert, \bigr \vert \log \frac{1-P(\hat{y}=1 \mid a=i)}{1-P(\hat{y}=1\mid a=j)} \bigr \vert \Bigr)
> > > $$

---

> > > > ### Comment · Reviewer_KRtS · 2023-11-23
> > > > **Thank you for detailed reponse**
> > > >
> > > > Thank you for the additional responses. They address my concerns sufficiently.

---

### Official Review · Reviewer_J4SN · 2023-11-03

**Soundness:** 3 good
**Presentation:** 2 fair
**Contribution:** 3 good
**Rating:** 5
**Confidence:** 3

**Summary:**

This work proposes a data augmentation technique to improve fairness, based on adversarial perturbations in the latent space. This technique leverages a covariance based constraint to to generate features which are maximally separated i.e. biased - followed by a training procedure to minimize the distance between the perturbed and original features - hoping to enforce some “invariance” over the true features i.e. not relying on the spurious / sensitive features. Experiments are demonstrated across multiple fairness datasets and vis-a-vis different methods, demonstrating the competitive fairness-accuracy tradeoff of the proposed method.

**Strengths:**

The paper addresses an important challenge of fairness by leveraging an adversarial latent space approach - which is interesting as most previous works demonstrate the opposite i.e. adversarial robustness maybe at odds with fairness. Further,  it is appreciated that detailed analysis is also shown on a synthetic dataset to better understand the hyperplane rotation i.e. how the boundary is encouraged to be changed. For the experiments, comparison is performed with many other methods.

**Weaknesses:**

Please see below:
1.  It is is unclear how this approach could be better than simple reweighting of the demographic groups, given that the knowledge is already available. Why would one like to leverage the adversarial data augmentation?
2. It is hard to infer the experimental conclusions from the many plots which are represented - is there any reason for not observing a non decreasing or non increasing trend in most of the experiments?
3. There seems to be no comparison with baselines such as reweighting and using only ERM features (no data augmentation) - how does the performance compare?
4. The motivation for choosing a covariance based approach is unclear - is there any specific intuition behind this choice?
5. Overall, the conciseness and coherence of the paper could be improved - e.g. :During the adversarial training, we ensure that the semantic essence of the perturbed features is preserved “ -> semantic essence could be replaced with a more technical term w.r.t the true invariant / sensitive attributes.

**Questions:**

Questions:
Apart from the questions mentioned in the above section:
1. Is there any underlying assumption on the structure of the space? Perhaps, it would be helpful to have a causal model explaining the structure of the input space (e.g. to depict the spurious correlation between Y and A).
2. Does this method scale beyond binary Y and A?
3. Could you compare your intuitions with the following adversarial group robustness technique?  Paranjape, Bhargavi, et al. "AGRO: Adversarial Discovery of Error-prone groups for Robust Optimization." arXiv preprint arXiv:2212.00921 (2022)

Suggestions:
1. I would recommend adding a concise summary of the plots (e.g. a table) to show the demonstrated improvements.
2. "Hyperplane rotation" may be hard to understand in the first read, could we replaced with a more common term like decision boundary.

---

> ### Author Response · Authors · 2023-11-18
>
> ### The benefit of the proposed method compared to data reweighting.
> Existing data reweighting methods [1], and [2] assign weights for all samples
> according to their importance, and re-train the entire model using the weighted data. On the other hand, ALFA freezes the encoder and fine-tunes the last layer only using the augmented data. Therefore, ALFA is more effective than existing reweighing methods in terms of time and memory cost. Moreover, according to [3] data reweighting always overfit, so there's no guarantee that the worst-group test performance to be better than ERM. In contrast, we directly consider both the overestimated (high false positive rate) and underestimated group (high false negative rate), seeking a fair misclassification rate for them.
>
>
> ### Description of the Pareto Frontier
> Thanks for pointing out. We will revise the explanation for the figure. In detail, the Pareto Frontier shows the trade-off between accuracy and fairness metric, while the black solid line shows the optimal trade-off. The closer a point to the black line, the better the accuracy-fairness trade-off is. In most cases, ours is located in the upper-right of the figure, which means it can achieve fairness while minimizing the compromising of accuracy.
>
> In fact, most of the existing frameworks shown in the figure are designed to improve fairness while maintaining accuracy. And only a few cases the fairness frameworks can improve accuracy too. It is hard to observe either decreasing or increasing accuracy in the experimental result. Moreover, we exclude some experimental results if they do not converge well, or are worse than the base rate.
>
> ### Comparison to reweighting and baseline
> We mainly focus on augmentation-based methods or some methods conducted in the latent space. However, as the reviewer asked, we can further compare the baseline and reweighting methods. In particular, we implement the state-of-the-art reweighting methods [2]. The revised Pareto Frontier, Figures 2 and 3 in revised paper's PDF file shows that ALFA outperforms the baseline and reweighting method. Moreover, the improvement in [2] is not consistent, whereas ALFA shows promising improvement.

---

> > ### Author Response · Authors · 2023-11-18
> >
> > ### Why covariance-based objective function?
> >
> > Other objective functions regarding fairness may be applicable to our framework such as [4]. We elaborate on the detail of the convex fairness constraint in the rebuttal PDF file and report the experimental results in the table below by comparing the baseline, the covariance-base fairness attack (suggested in the paper), and the convex fairness attack. The experiment shows that our method can adopt any type of fairness constraint during the attacking step.
> >
> > The reason we chose the covariance-based [5] is because it is simple to implement, and provable for effectively attack on fairness as shown in Theorem 3.2. To sum up, we can empirically show that other fairness constraints work well. However, we chose the covariance-based method because it is mathematically solid.
> >
> >
> > | Adult | Accuracy |  | $\Delta DP$ |  | $\Delta EOd$ |  |
> > | --- | --- | --- | --- | --- | --- | --- |
> > | Model | mean | std. | mean | std. | mean | std. |
> > | Logistic (Baseline) | 0.8465|0.0006|0.1798|0.0012|0.1891|0.0062|
> > | Logistic + ALFA (covariance) |0.8164|0.0013|0.1433|0.0035|0.1081|0.0067|
> > | Logistic + ALFA (convex) | 0.8227|0.0026|0.0852|0.0078|0.1547|0.0133|
> > | MLP (Baseline) |0.8458|0.0022|0.1835|0.0191|0.1781|0.0429|
> > | MLP + ALFA (covariance) | 0.8207|0.0019|0.1386|0.0045|0.0803|0.0062|
> > | MLP + ALFA (convex) | 0.8324|0.0031|0.1400|0.0166|0.0904|0.0184|
> >
> > | COMPAS | Accuracy |  | $\Delta DP$ |  | $\Delta EOd$ |  |
> > | --- | --- | --- | --- | --- | --- | --- |
> > | Model | mean | std. | mean | std. | mean | std. |
> > | Logistic (Baseline) |0.6575 |0.0011|0.2793|0.0103|0.5442|0.0214|
> > | Logistic + ALFA (covariance) | 0.6683|0.0039|0.0265|0.0138|0.0973|0.0344|
> > | Logistic + ALFA (convex) |  0.674|0.0034|0.0470|0.0180|0.1444|0.0379|
> > | MLP (Baseline) | 0.6695|0.0052|0.1173|0.0132|0.1845|0.0284|
> > | MLP + ALFA (covariance) | 0.6665|0.0029|0.0195|0.0054|0.0918|0.0111|
> > | MLP + ALFA (convex) | 0.6624|0.0010|0.0130|0.0075|0.0738|0.0150|
> >
> > | German | Accuracy |  | $\Delta DP$ |  | $\Delta EOd$ |  |
> > | --- | --- | --- | --- | --- | --- | --- |
> > | Model | mean | std. | mean | std. | mean | std. |
> > | Logistic (Baseline) | 0.7280|0.0283|0.1030|0.0776|0.3101|0.1440|
> > | Logistic + ALFA (covariance) | 0.7455|0.0159|0.0208|0.0212|0.1250|0.0512|
> > | Logistic + ALFA (convex) |  0.7410|0.0130|0.0240|0.0179|0.1030|0.0360|
> > | MLP (Baseline) | 0.7615|0.0032|0.0220|0.0082|0.2171|0.0111|
> > | MLP + ALFA (covariance) | 0.7405|0.0072|0.0101|0.0048|0.1794|0.0051
> > | MLP + ALFA (convex) | 0.7575|0.0087|0.0181|0.0120|0.1960|0.0079|
> >
> > | Drug | Accuracy |  | $\Delta DP$ |  | $\Delta EOd$ |  |
> > | --- | --- | --- | --- | --- | --- | --- |
> > | Model | mean | std. | mean | std. | mean | std. |
> > | Logistic (Baseline) | 0.6682|0.0133|0.2848|0.0397|0.4859|0.0827|
> > | Logistic + ALFA (covariance) | 0.6528|0.0081|0.0358|0.0177|0.1439|0.0648|
> > | Logistic + ALFA (convex) |  0.6509|0.0072|0.0596|0.0198|0.1284|0.0286|
> > | MLP (Baseline) | 0.6501|0.0075|0.2426|0.0247|0.4159|0.0489|
> > | MLP + ALFA (covariance) | 0.6440|0.0047|0.0948|0.0208|0.1135|0.0424|
> > | MLP + ALFA (convex) | 0.6329|0.0173|0.1002|0.0826|0.1955|0.0956|
> >
> >
> > ### Coherence of the paper
> > Thanks for pointing out. We will thoroughly revise paragraphs to be more readable and understandable, especially for the points suggested such as 'semantic essence', the readability of plots, and 'hyperplane rotation'.
> >
> > ### Causal structure assumption
> > Although we don't explicitly use the causal structure, we assume the causal structure like this figure, the sensitive attribute ($A$) implicitly affects other features ($X$), and the final classifier learns the latent feature $Z$ produced by an encoder utilizing from $A$ and $X$. But ALFA mitigates bias in $Z$, producing $Z^\prime$, which is independent of A. **We present the causal structure depicted in Appendix L of the revised paper's PDF file.**
> >
> > ### Beyond binary $Y$ and $A$
> > Please see the global response, addressing more general scenarios beyond binary classification with binary-sensitive attributes.

---

> > > ### Author Response · Authors · 2023-11-18
> > >
> > > ### Comparison to AGRO
> > > The rough ideas of 'finding unfair regions' in our method and 'identifying error-prone groups' in AGRO [6] are similar. However, the main differences between them are 1) computational cost, and 2) the objective of adversarial training.
> > > - First, AGRO suggested an end-to-end framework to mitigate spurious correlation, while ours is a fine-tuning method, which doesn't require the encoder doesn't have to be trained. As only the final classifier is trained in ALFA, our method is more effective.
> > > - Second, adversarial training in AGRO focuses on discovering the error-prone group which is not accurate though the group information is unknown. After the inference of the 'worst-group', the classifier aims to minimize the error of the worst group. In a word, AGRO is an improved version of G-DRO [7] which requires group information. In terms of the usage of group information, AGRO is a prominent methodology that can automatically detect the error-prone group.
> > >
> > >     However, our proposed method considers both the underestimated and overestimated groups, where one of them could be the worst group in AGRO. For example, if a privileged group with a negative label ($y=0,a=1$) has the highest misclassification rate, it is an overestimated group and an error-prone (worst) group too. On the other hand, if an unprivileged group with a positive label ($y=1,a=0$) has a high misclassification rate but not as much as in the worst-group, it is an underestimated group but not defined as an error-prone group in AGRO. To sum up, we address more diverse cases. As shown in Figure 2 in the rebuttal PDF files, both overestimated and underestimated groups are considered in ALFA. In detail, given the demographic information, ALFA detects 'unfair regions' which implies either disproportionately overestimated or underestimated, and directly recovers the region using a fairness attack as shown in Figure 1 in the manuscript.
> > >
> > >     Although ours considers both overestimated and underestimated groups while AGRO considers underestimated groups only, the idea of AGRO seeking the error-prone group without the group information is great. It could be a future direction of our fairness attack method operating without the sensitive attribute.
> > >
> > > - Third, the objectives of AGRO and ours are different. As AGRO addresses the spurious correlation of the model, it is interested in the worst-group accuracy only. In contrast, as our objective is to make a fair classifier, we are interested in the balanced true and false positive rates across the demographic group. Although they share the interest in mitigating bias in the model, the difference in the objective and evaluation metric provokes different thoughts about the subgroup we take care of.
> > >
> > > [1] Junyi Chai and Xiaoqian Wang. Fairness with adaptive weights. In International Conference on Machine Learning, pp. 2853-2866. PMLR, 2022.
> > >
> > > [2] Peizhao Li and Hongfu Liu. Achieving fairness at no utility cost via data reweighing with influence. In International Conference on Machine Learning, pp. 12917-12930. PMLR, 2022.
> > >
> > > [3] Runtian Zhai et al. “Understanding Overfitting in Reweighting Algorithms for Worst-group Performance”. In:
> > > (2021)
> > >
> > > [4] Yongkai Wu, Lu Zhang, and Xintao Wu. On convexity and bounds of fairness-aware classification. In The World Wide Web Conference, pp. 3356-3362, 2019.
> > >
> > > [5] Muhammad Bilal Zafar, Isabel Valera, Manuel Gomez Rogriguez, and Krishna P Gummadi. Fairness constraints: Mechanisms for fair classification. In Artificial intelligence and statistics, pp.962-970. PMLR, 2017.
> > >
> > > [6] Paranjape, Bhargavi, et al. "AGRO: Adversarial Discovery of Error-prone groups for Robust Optimization." arXiv preprint arXiv:2212.00921 (2022)
> > >
> > > [7] Shiori Sagawa, Pang Wei Koh, Tatsunori B Hashimoto, and Percy Liang. Distributionally robust neural networks for group shifts: On the importance of regularization for worst-case generalization. arXiv preprint arXiv:1911.08731, 2019.

---

### Official Review · Reviewer_VbNu · 2023-11-06

**Soundness:** 2 fair
**Presentation:** 2 fair
**Contribution:** 2 fair
**Rating:** 5
**Confidence:** 4

**Summary:**

The paper considers fairness in binary classification with binary protected features. In particular, focusing on DP and EOdds notions of group-level fairness, the paper proposes adversarial latent feature augmentation (ALFA), which considers the adversarial attack together with data augmentation. Experimental results are also provided.

**Strengths:**

The strength of the paper comes from the attempt to address adversarial attack and (latent space) data augmentation at the same time, for certain group-level fairness notions.

**Weaknesses:**

The weakness of the paper comes from the relatively unclear presentation of the material, the worry of the limited scope of application, and therefore, the unclear takeaway message. It would be helpful if points in __Questions__ below can be clarified.

**Questions:**

__Q1__: regarding "data augmentation in latent space"

The paper sets a goal of addressing adversarial attack and data augmentation at the same time for fair classification. However, it is not clear to me what exactly is being considered in the "latent space". The theoretical derivation closely follows the previous work Zafar et al., (2017), but with a different loss function (Equation 1). The "latent feature" $\mathbf{z}$ is utilized to attack the classifier. What is the relation between the original input features and the latent feature after attack? I am having difficulty understanding the motivation behind introducing an alteration in latent feature space for the purpose of both augmenting data and improving fairness.

__Q2__: what is "unfair region"

In Figure 1a, "unfair regions" are highlighted near the decision boundary (hyperplane). While I understand the fact that the groups are imbalanced, and that the proposed approach yields a "rotated" decision boundary, I am not sure how to parse the goal of finding a different slope (i.e., rotating) for the hyperplane in terms of data augmentation for fair classification. Please clarify.

__Q3__: more general scenarios beyond binary classification with binary protected feature

Can the proposed framework handle more complicated settings in practical scenarios? What is the takeaway message the paper would like to convey?

---

> ### Author Response · Authors · 2023-11-18
>
> ### The details about "data augmentation in latent space" and "unfair region"
> In the binary classification scenario, a model might be biasedly trained due to data imbalance in the demographic group, producing inconsistent misclassification rates across the subgroup, which we define as an 'unfair region' in the paper.
>
> ***Please see the Appendix K in the paper for the figures.***
> Here's an example of synthetic data explaining the concept of unfair region and how the decision boundary is rotated. We simply the binary classification task with a 2D Gaussian mixture data, consisting of two classes $y \in \\{0, 1\\}$ and two sensitive attributes, $A \in \\{0, 1\\}$ (indicating unprivileged and privileged groups.)
> \begin{align}
>     x \sim \begin{cases}
>         group1: \textbf{N} (\begin{bmatrix}
>            \mu \\\\
>            \mu
>          \end{bmatrix} , \sigma^2 )& \text{if} \\: y=1, a=1 \\\\
>         group2: \textbf{N} (\begin{bmatrix}
>            \mu \\\\
>            \mu^\prime
>          \end{bmatrix} , \sigma^2)& \text{if} \\: y=0, a=1 \\\\
>         group3: \textbf{N} (\begin{bmatrix}
>            0 \\
>         \mu
>          \end{bmatrix} ,(K\sigma)^2) & \text{if}\\: y=1, a=0 \\\\
>         group4: \textbf{N} (\begin{bmatrix}
>            0 \\\\
>            0
>          \end{bmatrix} , (K\sigma)^2)& \text{if}\\: y=0, a=0
>     \end{cases}
> \end{align}
> where $\mu^\prime =r\mu, 0<r<1$ and $K>1$, where the number of samples in each group is $N_1 : N_2: N_3: N_4$.  We arbitrarily set $K=3$, $r=0.7$, $\mu = 1$, $N_1 = N_2 = 100$, and $N_3=N_4=400$.
>
> From the given synthetic data, we have a decision boundary like Figure 1 in the rebuttal pdf. Due to the imbalance in the dataset, the subgroup $a=1,y=0$ is overestimated as label $y=1$, and the subgroup $a=0, y=1$ is underestimated as label $y=0$. The difference in misclassification rate is shown in Figure 2. We define them as 'unfair regions' where the misclassification rate is disproportionately high.
>
> Our fairness attack shifts the data distribution in a more biased way. In the synthetic data, as the demographic group $a=1$ is privileged to be predicted as $y=1$, group $a=1$ shifts towards the positive area, while group $a=0$ towards the negative area. Although this perturbation is counter-intuitive, the perturbed data directly recover the unfair region, producing a new decision boundary.
>
> Figure 2 shows that when we vary the amount of perturbation $\delta$ from 0 to 0.2, the misclassification rate for each group changes accordingly. As the fairness evaluation metric $\Delta EOd$ is defined as the summation of the True Positive Rate (TPR) gap and False Positive Rate (FPR) gap between the demographic groups, we plot the TPR gap, FPR gap, $\Delta EOd$, and the overall misclassification rate. The lower subfigure in Figure 2 shows that the TPR gap and FPR gap drop significantly achieving fairness while maintaining the overall misclassification rate.
>
> ### More general scenarios beyond binary labels and sensitive attributes
> Please see the global response, addressing more general scenarios beyond binary classification with binary-sensitive attributes.

---

> > ### Comment · Reviewer_VbNu · 2023-11-21
> > **Follow-up Questions**
> >
> > Thank authors for the additional experimental results and responses. My original questions and concerns are still not resolved.
> >
> > The term "(un)fair" and "bias" are heavily overloaded in the paper and the response. The data imbalance is irrelevant to what predictor one would like to use, and what fairness notion on prediction one would like to apply. By "fairness attack shifts the data distribution in a more biased way", the data imbalance (which may be problematic in itself and may be not) is mixed together with the DP violation. What exactly is the goal of ALFA? Is it intended to be a general strategy of data augmentation for different fairness notions? Or, is it specifically designed to deal with group-level DP notion, which happen to be closely related to "imbalance" in prediction $\hat{Y}$ (instead of the data $Y$)?
> >
> > In addition, I am still having difficulty parsing the connection between "data augmentation in latent space" and the characterization of "unfair region". While I understand the highlighted region in the figure corresponds to "unfair region", the characterization is too general to be informative. What is the "unfair region" in cases beyond binary classifications? What is the relation between rotating (in a general sense) the classification hyperplane and augmenting the data in latent space?

---

> > > ### Author Response · Authors · 2023-11-22
> > >
> > > Thank you for taking the time to respond to our rebuttal.
> > >
> > > We are focusing on the correlation between the sensitive attribute $A$ and the prediction $\hat{Y}$. We denote a group of misclassified instances causing high correlation as an unfair region for each sensitive attribute. Unfair region invokes high $\Delta DP$ and $\Delta EOd$ since it gives a higher chance to be predicted as positive to the privileged group (higher FPR) and less chance to the unprivileged group (higher FNR).
> > >
> > > ALFA's goal is to improve both $\Delta DP$ and $\Delta EOd$ by mitigating the correlation represented by the unfair regions. In detail, as the unfair regions are overestimated or underestimated, correcting the unfair regions makes the classifier fairer in terms of both $\Delta DP$ and $\Delta EOd$. Therefore, ALFA is a general strategy of data augmentation for different fairness notions, even beyond the binary scenario.
> > >
> > > In the case of multi-class classification or multiple sensitive attributes, it is hard to define the correlation between $A$ and $\hat{Y}$. However, we can define the unfair region as ‘any subgroup having a higher misclassification rate’. ALFA's strategy is to generate samples in unfair regions having correct labels. Although this data augmentation doesn't necessarily 'rotate' the decision boundaries in the case of multiple labels or protected attributes, the augmented latent features lead the classifier to reduce the misclassification rate in the unfair region. ALFA is more focused on the data augmentation in the unfair region, but we can observe the decision boundary rotation in some cases as a result of the data augmentation.
> > >
> > > Moreover, the reason we consider the unfair regions and data augmentation on latent space is because of the linear separability assumption which makes defining unfair regions easier. Linear separability is a widespread assumption, and deep neural networks transform the input data into well-separated latent features. In contrast, the transparency data augmentation in the input space is not obvious because of the non-linearity of the deep models. Therefore, defining unfair regions and recovering them using data augmentation is more straightforward in the latent space.

---

### Official Review · Reviewer_qEGE · 2023-11-09

**Soundness:** 2 fair
**Presentation:** 2 fair
**Contribution:** 2 fair
**Rating:** 6
**Confidence:** 2

**Summary:**

This paper proposes ALFA as a new method to enhance the fairness of classification models. It introduces the concept of adversarial perturbation in latent variable space, and demonstrates that fine-tuning on these perturbed features can lead to rotated decision boundaries, covering unfair regions. Experiments verify the effectiveness of ALFA in achieving group fairness while maintaining accuracy.

**Strengths:**

*Quality*
- This paper proposed a new approach to enhance fairness in machine learning models was proposed. This approach effectively merged data augmentation and adversarial attacks in the latent space to promote fairness. It revealed a counter-intuitive relationship between adversarial attacks and enhanced model fairness upon training with the resultant perturbed latent features.
- The theoretical and experimental results are correct to the best of my knowledge. Theoretical proof showed that the approach could generate biased feature perturbation, and experiments validated that training with adversarial features can improve fairness.

*Relevance*

The topic of fairness is important in machine learning.

**Weaknesses:**

*Novelty*

- Although the proposed method is new, the idea of using adversarial training and data augmentation for fairness enhancement is not novel. Adversarial training has been widely used to improve model robustness and generalization, while data augmentation is a common technique to enrich training datasets and reduce overfitting. Integrating these two concepts to address fairness issues has already been studied. Thus, further research is needed to explore the boundaries and limitations of this approach, as well as its applications in different domains.

*Algorithmic Analysis*

- One of the key aspects of ALFA's design is its ability to generate biased feature perturbations during training. This is achieved by introducing a novel loss function that encourages the model to focus on features that are particularly sensitive to small perturbations in the input space. While this approach can effectively rotate the decision boundary to cover previously unfavorable regions, it may also introduce computational complexity and training time challenges. ALFA's computational requirements are not explicitly stated in the article, but it's essential to consider them in order to evaluate its practicality for large-scale datasets and complex models.

- Additionally, while ALFA may improve fairness measures for certain groups, it's important to consider its overall impact on model stability and robustness. Adversarial loss can make training process vulnerable to small perturbations in the input space, potentially leading to increased numerical instability. This issue is not addressed in the article, and further research is needed to investigate these potential trade-offs when employing ALFA in practice.

- Moreover, while the experimental results provided in the article are encouraging, they only cover a limited number of datasets and model types. It remains to be seen how ALFA performs on other datasets that may exhibit different characteristics and challenges. Extensive validation and comparison with other state-of-the-art methods are necessary to assess ALFA's overall effectiveness and superiority.

**Questions:**

- To what extent is the proposed ALFA method innovative? Is it based on novel ideas or techniques that have not been previously explored in the field of fairness-enhancing machine learning? ALFA's algorithm design incorporates adversarial training and data augmentation. What are the novel aspects of this integration, and how does it lead to improved fairness?

- The use of adversarial training can potentially lead to instability. How does ALFA address this issue, and what measures are taken to ensure algorithm stability during training and deployment?

- ALFA's design involves a novel loss function that encourages the model to focus on features that are particularly sensitive to small perturbations in the input space. How does this loss function compare to traditional loss functions used in fairness-enhancing machine learning?

---

> ### Author Response · Authors · 2023-11-18
>
> ### Novelty of the proposed method
>
> Existing data augmentation methods are conducted on the input space. Although some of them utilize adversarial training to generate augmented data, the transparency and efficacy of transformation in the input space are not obvious because of the non-linearity of the deep models. Moreover, the objectives of adversarial training in the existing methods are different from ours. In detail, FAAP [1] adopts adversarial training to align the data to the auxiliary decision boundary predicting sensitive attributes rather than labels. AntiDRO [2] generates antidote data produced by Distributionally Robust Optimization (DRO) to select the worst-performing data. The generated data is used as data augmentation while having analogous features from the original dataset but different sensitive attributes. However, AntiDRO is designed only for individual fairness, and training on these antidote data cannot improve group fairness. In contrast, our combination of data augmentation and adversarial training is not trivial. In detail, we define 'unfair regions' where a demographic group is disproportionately overestimated (high false positive rate) and underestimated data (high false negative rate). The fairness attack automatically finds the unfair regions and directly makes the perturbed data recover the unfair regions. In this step, the perturbed data itself could be a biased dataset, but we utilize them as data augmentation which is a counter-intuitive way to mitigate bias by making the misclassification rates for each subgroup fair.
>
>
>
> ###  Computational complexity and training time
>
> When it comes to computational complexity, ALFA requires the same memory space as the original training during the attacking step. In the fine-tuning step, as it stores perturbed latent features as a matrix form and is directly applied to the final classifier, only the final classifier needs the additional computational cost for the single latent feature matrix $Z^\prime \in \mathbb{R}^{N \times d}$ where $N$ is the number of samples and $d$ is the dimension of the latent feature.
> For example, let's assume we use $L$ layers of MLP, and each layer has the same dimension $d$ for simplicity. Then, the computational complexity is $\mathcal{O}(pdN + (L-1) d^2N + dN) )$ where $p$ is the number of features of the input. During the attacking step, we freeze the encoder and classifier, only computing the covariance and the perturbation update, where each of them takes $\mathcal{O}(dN)$ complexity. Moreover, during the fine-tuning step, it also takes complexity of $\mathcal{O}(dN)$. Therefore, the additional computational complexity for an iteration is $\mathcal{O}(3dN)$ including covariance calculation, perturbation update, and fine-tuning, which is significantly light compared to the ERM.
>
>
> ### Stability of the proposed method
>
> In typical adversarial training, the min-max optimization or bi-level (alternative) optimization may not be stable. However, it doesn't happen in ALFA since it requires a one-time attacking step to maximize the fairness constraint, and the augmented data is used in the training to minimize binary cross-entropy loss. Therefore, our framework has the same stability as traditional empirical risk minimization (ERM).
>
> ### Extension to more complex dataset and model
> Please see the global response addressing more complex situations.

---

> > ### Author Response · Authors · 2023-11-18
> >
> > ### Difference from existing fairness loss function
> >
> > Unlike the existing works, ALFA shows a counter-intuitive usage of fairness constraint, maximizing it rather than minimizing it, which is a novel perspective. We proved that maximizing the fairness constraint effectively attacks the $\Delta DP$ and $\Delta EOd$.  However, our contribution is not limited to the usage of covariance-based fairness constraints [3].  Indeed, any type of fairness constraint can be used in the attacking step. For example, we can adopt a convex fairness constraint [4]. We elaborate on the detail of the convex fairness constraint in the rebuttal PDF file and report the experimental results in the table below by comparing the baseline, the covariance-base fairness attack (suggested in the paper), and the convex fairness attack. The experiment shows that our method can adopt any type of fairness constraint during the attacking step.
> >
> >
> > | Adult | Accuracy |  | $\Delta DP$ |  | $\Delta EOd$ |  |
> > | --- | --- | --- | --- | --- | --- | --- |
> > | Model | mean | std. | mean | std. | mean | std. |
> > | Logistic (Baseline) | 0.8465|0.0006|0.1798|0.0012|0.1891|0.0062|
> > | Logistic + ALFA (covariance) |0.8164|0.0013|0.1433|0.0035|0.1081|0.0067|
> > | Logistic + ALFA (convex) | 0.8227|0.0026|0.0852|0.0078|0.1547|0.0133|
> > | MLP (Baseline) |0.8458|0.0022|0.1835|0.0191|0.1781|0.0429|
> > | MLP + ALFA (covariance) | 0.8207|0.0019|0.1386|0.0045|0.0803|0.0062|
> > | MLP + ALFA (convex) | 0.8324|0.0031|0.1400|0.0166|0.0904|0.0184|
> >
> > | COMPAS | Accuracy |  | $\Delta DP$ |  | $\Delta EOd$ |  |
> > | --- | --- | --- | --- | --- | --- | --- |
> > | Model | mean | std. | mean | std. | mean | std. |
> > | Logistic (Baseline) |0.6575 |0.0011|0.2793|0.0103|0.5442|0.0214|
> > | Logistic + ALFA (covariance) | 0.6683|0.0039|0.0265|0.0138|0.0973|0.0344|
> > | Logistic + ALFA (convex) |  0.674|0.0034|0.0470|0.0180|0.1444|0.0379|
> > | MLP (Baseline) | 0.6695|0.0052|0.1173|0.0132|0.1845|0.0284|
> > | MLP + ALFA (covariance) | 0.6665|0.0029|0.0195|0.0054|0.0918|0.0111|
> > | MLP + ALFA (convex) | 0.6624|0.0010|0.0130|0.0075|0.0738|0.0150|
> >
> > | German | Accuracy |  | $\Delta DP$ |  | $\Delta EOd$ |  |
> > | --- | --- | --- | --- | --- | --- | --- |
> > | Model | mean | std. | mean | std. | mean | std. |
> > | Logistic (Baseline) | 0.7280|0.0283|0.1030|0.0776|0.3101|0.1440|
> > | Logistic + ALFA (covariance) | 0.7455|0.0159|0.0208|0.0212|0.1250|0.0512|
> > | Logistic + ALFA (convex) |  0.7410|0.0130|0.0240|0.0179|0.1030|0.0360|
> > | MLP (Baseline) | 0.7615|0.0032|0.0220|0.0082|0.2171|0.0111|
> > | MLP + ALFA (covariance) | 0.7405|0.0072|0.0101|0.0048|0.1794|0.0051
> > | MLP + ALFA (convex) | 0.7575|0.0087|0.0181|0.0120|0.1960|0.0079|
> >
> > | Drug | Accuracy |  | $\Delta DP$ |  | $\Delta EOd$ |  |
> > | --- | --- | --- | --- | --- | --- | --- |
> > | Model | mean | std. | mean | std. | mean | std. |
> > | Logistic (Baseline) | 0.6682|0.0133|0.2848|0.0397|0.4859|0.0827|
> > | Logistic + ALFA (covariance) | 0.6528|0.0081|0.0358|0.0177|0.1439|0.0648|
> > | Logistic + ALFA (convex) |  0.6509|0.0072|0.0596|0.0198|0.1284|0.0286|
> > | MLP (Baseline) | 0.6501|0.0075|0.2426|0.0247|0.4159|0.0489|
> > | MLP + ALFA (covariance) | 0.6440|0.0047|0.0948|0.0208|0.1135|0.0424|
> > | MLP + ALFA (convex) | 0.6329|0.0173|0.1002|0.0826|0.1955|0.0956|
> >
> > [1] Zhibo Wang, Xiaowei Dong, Henry Xue, Zhifei Zhang, Weifeng Chiu, Tao Wei, and Kui Ren. Fairness-aware adversarial perturbation towards bias mitigation for deployed deep models. In Proceedings of the IEEE/CVF Conference on Computer Vision and Pattern Recognition, pp. 10379-10388, 2022b
> >
> > [2] Peizhao Li, Ethan Xia, and Hongfu Liu. Learning antidote data to individual unfairness. In International Conference on Machine Learning, pp. 20168-20181. PMLR, 2023.
> >
> > [3] Muhammad Bilal Zafar, Isabel Valera, Manuel Gomez Rogriguez, and Krishna P Gummadi. Fairness constraints: Mechanisms for fair classification. In Artificial intelligence and statistics, pp.962-970. PMLR, 2017.
> >
> > [4] Yongkai Wu, Lu Zhang, and Xintao Wu. On convexity and bounds of fairness-aware classification. In The World Wide Web Conference, pp. 3356-3362, 2019.

---

> > > ### Comment · Reviewer_qEGE · 2023-11-22
> > >
> > > Thanks for the authors feedback. I do not have further questions.

---

### Official Review · Reviewer_92u3 · 2023-11-13

**Soundness:** 2 fair
**Presentation:** 2 fair
**Contribution:** 2 fair
**Rating:** 6
**Confidence:** 3

**Summary:**

In this paper, the authors propose an adversarial data augmentation technique based on maximizing the covariance of sensitive attributes and signed distance from the decision boundary. They then incorporate the augmented samples into the training of the classifier layer. Overall, the authors demonstrate that their method improves fairness metrics over the considered baselines.

**Strengths:**

- The proposed method is simple and effective
- The presentation of the paper is good.

**Weaknesses:**

- The paper considers fine-tuning only the classifier part of the network (i.e., the last layer) and freezes the encoder part of the network. While this method works for simple tabular-like datasets, I have concerns about how this method performs in the case of datasets with a large label space, like ImageNet.

- In general, the idea is not novel as the perturbation in the latent space is already explored by the earlier methods

**Questions:**

Please see the weakness section above and in addition to that:

- How is the perturbation $\epsilon$ set for different feature encoding layers in different ML models? This is an additional hyperparameter that needs to be carefully tuned. Moreover, the authors mention hyperparameters in Table 2 in the appendix. It would be fair if the authors mentioned the cost of hyperparameter tuning in terms of time for the proposed method compared to the baselines.

- How is the performance in datasets with large label space?

- What is the performance of the adversarial training by perturbing the input space instead of latent space with similar attack objective?

- In section 4.4.1, authors could quantitatively discuss the  improvements in terms of performance and fairness metrics. It is not clear to me from the graphs about the comparison of different methods.

---

> ### Author Response · Authors · 2023-11-18
>
> ### Extension to diverse data types and scenarios
> We adopt the fine-tuning strategy of freezing the encoder part, our experimental results show that this strategy effectively achieves fairness even for the image dataset (CelebA) and text dataset (Wiki), as shown in the manuscript and the global response.
>
> We also further research the usage of ALFA in the multi-class classification, as it can be conceptualized as multiple One-Vs-All binary classifications as presented in the global response. Though ALFA can address multi-class classification, the improvement is not significant, and the dataset used in the experiment is not large enough in terms of its size and label space.
>
> We summarize our future works here. First, we will find or construct a large dataset having fairness concerns since the existing benchmark datasets for fairness research are relatively small in terms of size and label space. As we have verified the effectiveness of finding and directly recovering unfair regions, we expect that ALFA can be applied to large label space. Second, we are interested in the long-tail classification scenario, where the label space is large and extremely imbalanced. Although this topic is not directly related to the fairness problem, we motivate this scenario because it might suffer from highly overestimated or underestimated classes, which our framework can address.
>
> ### Comparison to existing perturbation methods in latent space
> We have investigated the literature about latent perturbation methods for fairness and identified two existing works [1,2]. Both use GAN for the perturbation, and the direction of the perturbation is regarding the auxiliary decision boundary, produced by the sensitive attribute. In detail, [1] generates a perturbed sample having opposite sensitive attributes but the same target labels locating it in a symmetric point. Therefore, the role of perturbation is to make the number of samples and distribution similar across the subgroups. However, [1] is based on the strong assumption that each subgroup's data distribution should be symmetric about the decision boundaries. [2] also uses the sensitive attribute decision boundary and generates the perturbation towards that decision boundary. We show the conceptual image showing that this perturbation doesn't necessarily lead to the fair decision boundary, as shown in Figure 5(b) in the paper. Moreover, both [1] and [2] are designed only for the image dataset. In our paper, we empirically show that FAAP [2] produces poor results in tabular datasets as shown in Figure 2 and 3.
>
> **The novelty of our method can be summarized:**
> 1) the adaptiveness to diverse types of data,
> 2) defining unfair regions where a demographic group is disproportionately overestimated (high false positive rate) and underestimated (high false negative rate),
> 3) directly recovering the unfair regions using fairness attack which is counter-intuitive usage of fairness constraint, generating biased dataset to mitigate bias.
>
> Although we have identified only two existing methods, we are open to further discussion about the perturbation method in latent space for fairness. In particular, we will study further how to build interpretable latent perturbation for fairness.

---

> > ### Author Response · Authors · 2023-11-18
> >
> > ### Faster hyperparameter tuning
> >
> > The hyperparameter $\epsilon$ is applied to the perturbation for the latent feature before the last classifier, while the other layer's latent features are not involved in the perturbation.
> >
> > We agree with the reviewer that time consumption for hyperparameter tuning could be burdensome, thus in the following we suggest the following strategy to set hyperparameter for our method. First, the hyperparameter $\epsilon$ can be replaced with the average Euclidean distance between samples and the decision boundary since it is possible to use either $L2$ perturbation or $L_\infty$ perturbation interchangeably in our framework. When it comes to $\lambda$, the selection of $\lambda$ is significant to the accuracy-fairness trade-off since it works as the weight between the original and perturbed dataset. Therefore, we vary $\lambda$ and report the 'line' of the performances in the Pareto Frontier as well as varying $\alpha$ in Eq.(5). We suggest using $\lambda=0.5$ since it empirically shows the best accuracy-fairness trade-off for all datasets.
> >
> >
> > Still, the hyperparameter tuning is needed for $\alpha$. Here we report the training time for experiments for each combination of hyperparameters. Each fine-tuning is conducted in 50 epochs. We measure the training time for a single run for baseline, and the average of 10 runs for fine-tuning.
> >
> > | Dataset | Baseline | Fine-tuning|
> > | --- | --- | --- |
> > | Adult | 40.046s| 37.018s |
> > | COMPAS |9.118s| 6.166s|
> > | German |3.934s |1.177s|
> > | Drug| 4.748s | 1.838s|
> >
> > Therefore, we can omit the hyperparameter $\epsilon$ and $\lambda$ by using $L2$ perturbation taking the mean Euclidean distance between the latent feature and the decision boundary as $\epsilon$ and setting $\lambda=0.5$ to make the original dataset and perturbed dataset equally contribute. However, even though only one hyperparameter, $\alpha$, is need to be tuned, the hyperparameter tunig itself takes less time compared to the original ERM because fine-tuning trains only the last layer of the model.
> >
> >
> > ### Perturbation in input space
> > In fact, as the logistic regression doesn't have an encoder, the fairness attack is conducted on the input space in this case. For other models, we observed that the same objective function replacing latent features to input features works comparably but slightly worse than the current version. Because our strategy, decision boundary rotation, is based on the assumption that the latent features are linearly separable, the perturbation on the input space doesn't effectively contribute to the rotation of the decision boundary due to the neural network's non-linear transformation.
> >
> >
> > ### Explanation of figure and quantitative discussion
> > We will revise the explanation for the figure. In detail, the Pareto Frontier shows the trade-off between accuracy and fairness metric, while the black solid line shows the optimal trade-off. The closer a point to the black line, the better the accuracy-fairness trade-off is. In most cases, ours is located in the upper-right of the figure, which means it can achieve fairness while minimizing the compromising of accuracy.
> >
> > For the fair quantitative comparison, we pick an experimental result when the accuracy is the highest among the points in the Pareto Frontier for each method. We report the table in Appendix G. The experimental results show that the fairness improvement is outstanding in most cases, achieving the best or second-best fairness score.
> >
> >
> > [1] Ramaswamy, Vikram V., Sunnie SY Kim, and Olga Russakovsky. "Fair attribute classification through latent space de-biasing." Proceedings of the IEEE/CVF conference on computer vision and pattern recognition. 2021.
> >
> > [2] Zhibo Wang, Xiaowei Dong, Henry Xue, Zhifei Zhang, Weifeng Chiu, Tao Wei, and Kui Ren. Fairness-aware adversarial perturbation towards bias mitigation for deployed deep models. In Proceedings of the IEEE/CVF Conference on Computer Vision and Pattern Recognition, pp. 10379-10388, 2022b

---

### Author Response · Authors · 2023-11-18

We thank the reviewers for suggesting the following experiments. We'll add the results for our final paper.

## General Scenarios beyond binary classification with binary sensitive attributes.
We explore further the effectiveness of the proposed method, showing the experimental results on multi-label classification, multiple sensitive attributes, multi-class classification, and extension to the more complex model (ResNet-like and Transformer-like architecture) and dataset (NLP).
### Multi-label Classification
We clarify that ALFA can be applied to the multi-label classification with binary-protected features as it can be seen in multiple binary classification scenarios having individual decision boundaries. In this case, the fairness loss is newly defined as covariance between a sensitive attribute and the mean of the signed distances, $L_{fair} = Cov(a, \frac{1}{T}\sum_{t=1}^T g_t (z_t+\delta_t))$ where $T$ is the number of targeted prediction.

Luckily, one of our datasets, the Drug Consumption dataset has multiple labels. To further investigate the feasibility of our framework for the multi-label classification, we conduct additional experiments on the Drug Consumption dataset [1] choosing four prediction goals, Cocaine, Benzodiazepine, Ketamine, and Magic Mushrooms while only Cocaine is considered as a prediction goal in the manuscript. The experimental result shows that ALFA effectively mitigates biases in the multi-label classification.

| Accuracy | Cocaine |  | Benzos |  | Ketamine |  | Mushrooms |  |
| --- | --- | --- | --- | --- | --- | --- | --- | --- |
| Model | mean | std. | mean | std. | mean | std. | mean | std. |
| Logistic Regression |  0.7057|0.0099|0.6689|0.0113|0.6989|0.0267|0.7223|0.0094|
| Logistic Regression + ALFA |  0.6816|0.0114|0.6643|0.0122|0.7505|0.0023|0.7307|0.0082|
| MLP |0.6802|0.0144|0.6527|0.0138|0.7551|0.0094|0.7053|0.0114|
| MLP + ALFA | 0.6701|0.0057|0.6138|0.0036|0.7343|0.0031|0.6587|0.0057|

| $\Delta DP$ | Cocaine |  | Benzos |  | Ketamine |  | Mushrooms |  |
| --- | --- | --- | --- | --- | --- | --- | --- | --- |
| Model | mean | std. | mean | std. | mean | std. | mean | std. |
| Logistic Regression | 0.2691|0.0232|0.3597|0.0298|0.2478|0.1140|0.4151|0.0372|
| Logistic Regression + ALFA | 0.0986|0.0289|0.2666|0.0424|0.0248|0.0070|0.3993|0.0425|
| MLP | 0.2183|0.0222|0.3179|0.0278|0.0903|0.1320|0.4072|0.0206|
| MLP + ALFA | 0.0760|0.0114|0.1808|0.0137|0.0368|0.0103|0.2384|0.0099|

| $\Delta EOd$ | Cocaine |  | Benzos |  | Ketamine |  | Mushrooms |  |
| --- | --- | --- | --- | --- | --- | --- | --- | --- |
| Model | mean | std. | mean | std. | mean | std. | mean | std. |
| Logistic Regression | 0.4411|0.0483|0.6448|0.0635|0.5184|0.2320|0.7096|0.0732|
| Logistic Regression + ALFA | 0.1234|0.0471|0.4498|0.0858|0.0689|0.0158|0.6621|0.0911|
| MLP | 0.3505|0.0449|0.5601|0.0597|0.2492|0.0385|0.6912|0.0441|
| MLP + ALFA | 0.0963|0.0249|0.2971|0.0193|0.1215|0.0153|0.3628|0.0185|


### Multi-protected features

In the binary classification with multi-protected features, the Differential Fairness (DF) is measured by binarization of each multi-protected features. For example, [2] defined DF
$$
 DF = \max_{i,j \in S} \Bigl(\max(\bigl \vert \log \frac{P(y=1 \vert a=i)}{P(y=1\vert a=j)}\bigl \vert, \bigr \vert \log \frac{P(y=1 \vert a=i)}{P(y=1\vert a=j)} \bigr \vert \Bigr)
$$
where $i,j \in S$, and $S$ denotes the set of the multiple sensitive attributes. Therefore, in the multi-protected feature case, we can define 'unfair region' by finding a particular sensitive attribute provoking the maximum mistreatment and reducing the misclassification rate of the unfair region as well as the binary sensitive attribute case.

For the multiple sensitive attribute setting, we adopt COMPAS dataset and MEPS [3] dataset. MEPS data consists  of 34,655 instances with 41 features(e.g. demographic information, health services records, costs, etc.) Among all the features, only 42 features are used. The sum of total medicare visiting is used as a binary target label. When the total number of visiting is greater or equal to 10, a patient is labeled as 1, otherwise 0. And 'race' is used as multiple sensitive attributes, 0 for White, 1 for Black, and 2 for others.

The experimental result shows that ALFA is also applicable to the multiple sensitive attributes scenario. (smaller DF is better.)

| COMPAS| Accuracy |  | DF |  |
| --- | --- | --- | --- | --- |
| Model | mean | std. | mean | std. |
| MLP | 0.6875|0.0048|1.7500|0.5794|
| MLP + ALFA | 0.6895|0.0023|1.3960|0.0892|

| MEPS| Accuracy |  | DF |  |
| --- | --- | --- | --- | --- |
| Model | mean | std. | mean | std. |
| MLP | 0.6208|0.0137|0.2900|0.0700|
| MLP + ALFA | 0.6860|0.0024|0.1985|0.0226|

---

> ### Author Response · Authors · 2023-11-18
>
> ### Multi-class classification
> For the multi-class classification, the decision boundaries are not linear, so our framework might not be directly applicable. However, multi-class classification can indeed be conceptualized as multiple binary classifications in a certain strategy called **One-Vs-All**. In this approach, for a problem with $N$ classes, we can create $N$ different binary classifiers. Each classifier is trained to distinguish between one of the classes and all other classes combined.
>
> As each classifier can be seen as a binary classification task, we can utilize ALFA for the multi-class classification scenario by detecting unfair regions and recovering the region by fairness attack.
> The evaluation metric for multi-class fairness takes maximum Demographic Parity across the classes [4]. In details, $$
>     \Delta DP_{\text{multi}} = \max_{k \in [K]} \bigl \vert  P(\hat{Y} = k \vert a=1 ) - P(\hat{Y} = k \vert a= 0 )\bigr \vert
> $$
> where $\hat{Y}$ is the predicted class, and $k\in[K]$ denotes each class $k$ in the multi-class classification.
>
> Among existing datasets for fairness research, Drug dataset can be used for multi-class classification. In fact, the original labels of the Drug dataset are multi-class settings, from 'CL0' to 'CL6' indicating the frequency of drug abuse. We have binarized them as 'never used' and 'ever used' regardless of the frequency in the main paper. However, for the multi-class classification setting, we adopt the original multi-class setting and report the mean accuracy and $\Delta DP_{\text{multi}}$ with MLP.
>
> | Drug-multi| Accuracy |  | $\Delta DP_{\text{multi}}$ |  |
> | --- | --- | --- | --- | --- |
> | Model | mean | std. | mean | std. |
> | MLP | 0.5196|0.0032|0.1930|0.0132|
> | MLP + ALFA | 0.4960|0.0219|0.1733|0.0287|
>
> ## Extension to complex cases
> ### Complex model
> In our current paper, we used four tabular datasets and one image dataset with Logistic Regression, Multi-Layer Perceptron (MLP), and ResNet-18, respectively for experiments. To verify the effectiveness of our framework on the tabular dataset, we extend the experiment using 1) a ResNet-like backbone, and 2) a Transformer-like backbone designed for tabular datasets both proposed in [5]. The revised Pareto Frontier, Figure 6 and 7 in Appendix I.4 in the revised manuscript show that ALFA is applicable to more complex model.
>
>
> ### Complex dataset
> Moreover, we further explore the adaptability of the proposed method to the Natural Language Processing (NLP) dataset. We utilize Wikipedia Talk Toxicity Prediction [6] which is a comprehensive collection aimed at identifying toxic content within discussion comments posted on Wikipedia's talk pages, produced by the Conversation AI project. In this context, toxicity is defined as content that may be perceived as "rude, disrespectful, or unreasonable." It consists of over 100,000 comments from the English Wikipedia, each meticulously annotated by crowd workers, as delineated in their associated research paper. A challenge presented by this dataset is the underrepresentation of comments addressing sensitive subjects such as sexuality, religion, gender identity, and race. In this paper, the existence of sexuality terms such as 'gay', 'lesbian', 'bisexual', 'homosexual', 'straight', and 'heterosexual' is used as the sensitive attribute, 1 for existing, and 0 for absence.
>
> For the NLP dataset, we use a pre-trained word embedding model, Glove [7] to produce input data and use MLP and LSTM network [8] for the experiments.
>
>
> | Wiki | Accuracy |  | $\Delta DP$ |  | $\Delta EOd$ |  |
> | --- | --- | --- | --- | --- | --- | --- |
> | Model | mean | std. | mean | std. | mean | std. |
> | MLP | 0.9345|0.0043|0.1735|0.0066|0.0734|0.0145|
> | MLP + ALFA | 0.9349|0.0002|0.1684|0.0019|0.0500|0.0049|
> | LSTM | 0.9294|0.0007|0.1922|0.0053|0.0806|0.0095|
> | LSTM + ALFA | 0.9280|0.1001|0.1872|0.0001|0.0607|0.0004|

---

> > ### Author Response · Authors · 2023-11-18
> >
> > [1] Dheeru Dua, Casey Graff, et al. Uci machine learning repository. 2017.
> >
> >
> > [2] James R Foulds, Rashidul Islam, Kamrun Naher Keya, and Shimei Pan. An intersectional definition of fairness. In 2020 IEEE 36th International Conference on Data Engineering (ICDE), pp. 1918–1921. IEEE, 2020
> >
> > [3] Rachel K. E. Bellamy et. al., AI Fairness 360: An extensible toolkit for detecting, understanding, and mitigating unwanted algorithmic bias, October 2018.
> >
> > [4] Christophe Denis, Romuald Elie, Mohamed Hebiri, and Franc ̧ois Hu. Fairness guarantee in multi-class classification. arXiv preprint arXiv:2109.13642, 2021
> >
> >
> >
> > [5] Gorishniy, Y., Rubachev, I., Khrulkov, V., & Babenko, A. (2021). Revisiting deep learning models for tabular data. Advances in Neural Information Processing Systems, 34, 18932-18943.
> >
> > [6] Nithum Thain, Lucas Dixon, and Ellery Wulczyn. “Wikipedia Talk Labels: Toxicity”. In: (Feb. 2017).doi:10.6084/m9.figshare.4563973.v2.
> >
> > [7] Pennington, Jeffrey, Richard Socher, and Christopher D. Manning. "Glove: Global vectors for word representation." Proceedings of the 2014 conference on empirical methods in natural language processing (EMNLP). 2014.
> >
> > [8] Sepp Hochreiter; Jürgen Schmidhuber (1997). "Long short-term memory". Neural Computation. 9 (8): 1735-1780

---

### Meta-Review · Area_Chair_dRcn · 2023-12-04

**Metareview:**

I have read all the materials of this paper including the manuscript, appendix, comments, and response. Based on collected information from all reviewers and my personal judgment, I can make the recommendation on this paper, *reject*. No objection from reviewers who participated in the internal discussion was raised against the reject recommendation.

**Research Question**

The authors focus on fairness learning in the classification scenarios.

**Motivations & Challenge Analysis**

When the authors analyze the limitations of existing methods, some statements are unclear or difficult to understand. For example, GAN-based perturbation might yield unsuitable generated features not aligning on the sensitive hyperplane.

**Philosophy**

The authors introduced a concept of unfair regions, but it is unclear on how to employ unfair regions to tackle the limitations of GAN-based perturbation.

**Unfair Regions**

There is no clear and formal definition of unfair regions. Moreover, this concept is disconnected with the proposed method.

**Technique**

The novelty is limited. There exist many studies using adversarial perturbations for performance improvements. In my eyes, the authors used several mature techniques to achieve fairness.

**Experiments**

1. The authors failed to demonstrate their method can tackle the limitations of GAN-based methods.

2. It is unclear how the authors tune the parameters on the validation set, since both utility and fairness are taken into consideration.

3. The performance of the proposed method is not consistent. Some analyses are needed.

4. More in-depth experiments are needed including the parameter analysis in Algorithm 1.

5. For Figure 5 (synthetic data), it should show the unfair regions and the generated samples.

**Presentation**

1. Figures are difficult to see and compare, where the legend font is too small.

2. Figure 1(b) is not informative. What I can tell from this figure is just the changed decision boundary. That is it and I do not know why. In another word, it does not help on understanding the proposed method. This figure might be moved to the experimental part.

3. Algorithm 1 should be moved to the position near Section 4.

**Justification For Why Not Higher Score:**

This paper is not self-standing and does not reach the bar of ICLR.

**Justification For Why Not Lower Score:**

N/A

---

### Decision · Program_Chairs · 2024-01-16

Reject